# Genetic tuning of retinal ganglion cell subtype identity to drive visual behavior

Marcos L. Aranda [1], Jacob D. Bhoi[1], Omar A. Payán Parra[1], Seul Ki Lee[1], Tomoko Yamada[1], Yue Yang[1] & Tiffany M. Schmidt [1,2] ✉

The distinct blend of molecular and cellular features that define neuronal subtype identity are central to shaping how individual subtypes impact animal behavior. The diversity of the mammalian nervous system is vast − the retina alone contains over 100 neuronal subtypes. Yet, the genetic processes giving rise to this stunning structural and functional diversity remain poorly understood. Here, we uncover a graded expression pattern of the transcription factor BRN3B that tunes and maintains multiple, subtype-defining transcriptional and morphophysiological features of the melanopsin-expressing, intrinsically photosensitive retinal ganglion cells (ipRGCs) in mice. Disruption of BRN3B expression levels causes the transcriptional and morphophysiological identity of ipRGC subtypes to begin to converge, leading to dysfunction in multiple ipRGC-dependent behaviors. These findings show that graded levels of a single transcription factor can tune a diverse array of features to shape neuronal identity and circuit function to drive behavior.

Light information from the retina is relayed by more than 40 types of retinal ganglion cells (RGCs) to more than 40 brain regions to mediate conscious visual perception and subconscious, non-image forming functions[1–5]. Each RGC type exhibits unique morphophysiological properties, projections, and roles in behavior[6–8]. However, the cellular mechanisms that give rise to this stunning diversity of RGC structure and function are unknown. The melanopsin (*Opn4*)-expressing, intrinsically photosensitive retinal ganglion cells (ipRGCs) are a class of RGC that play a key role driving both subconscious, non-image forming behaviors such as circadian photoentrainment and the pupillary light reflex as well as contrast sensitivity for conscious visual perception[9–14]. Different ipRGC subtypes (M1–M6) selectively drive these behaviors, with each subtype characterized by a distinct complement of morphophysiological properties and melanopsin expression levels[14]. Together, these subtype-defining features form a representative subsample of the broad array of morphophysiological features, projections, and roles in behavior that combine to define all RGC types, making ipRGCs a compelling system for elucidating the mechanisms that drive RGC diversity[3,8]. In this paper, we identify a graded expression pattern of a single transcription factor that is critical for tuning multiple, subtype-defining transcriptional and

morphophysiological features of ipRGC identity to shape their individual functions in diverse visual behaviors.

## Results

### BRN3B regulates ipRGC transcriptional identity

BRN3B (*Brn3b*/*Pou4f2*) is a transcription factor important for RGC specification in early embryonic development[15]. Notably, BRN3B expression is present in newly postmitotic ipRGCs and persists into adulthood, suggesting it may play yet unidentified roles in ipRGC development and function[16]. We therefore assessed how the removal of BRN3B from ipRGCs impacts gene regulation in these cells. Since ipRGCs constitute only ~0.01% of all retinal cells, we took advantage of the translating affinity purification (TRAP) approach to characterize ipRGC gene expression[17]. We crossed transgenic, *Opn4Cre* mice, which selectively express Cre-recombinase in ipRGCs, with mice conditionally expressing *Rpl22HA* in a Cre-dependent manner[18]. Purification and sequencing of mRNA bound to HA-*Rpl22* labeled ribosomes revealed 2326 transcripts enriched in ipRGCs compared to in the total retina (Supplementary Fig. 1). We next generated a BRN3B conditional knockout (Brn3bcKO) animal (*Opn4Cre*; *Brn3bcKOAP/cKOAP*) to eliminate BRN3B from postmitotic ipRGCs[15]. Developmentally, mRNA for *Brn3b*

[1]Department of Neurobiology, Northwestern University, Evanston, IL, USA. [2]Department of Ophthalmology, Feinberg, School of Medicine, Northwestern University, Chicago, IL, USA. ✉e-mail: tiffany.schmidt@northwestern.edu

and the mature RGC marker *Rbpms* are detectable at embryonic day (E) 12.5 in retinal sections, around the time that ipRGCs exit the cell cycle and become postmitotic (Supplementary Fig. 2)[19,20]. Importantly, *Opn4* mRNA expression is not detectable until E15.5, indicating that *Opn4*-dependent excision of the *Brn3b* gene occurs well after ipRGCs have become postmitotic (Supplementary Fig. 2). Using mRNA fluorescent in situ hybridization (FISH), we confirmed that the *Brn3b* expression found in control retinas is absent in ipRGCs of adult Brn3bcKO retinas (Supplementary Fig. 3). We then crossed the Brn3bcKO mouse line with the conditional *Rpl22^HA* line and performed TRAPseq using retinas from Brn3bcKO (*Opn4^Cre; Brn3b^cKOAP/cKOAP; Rpl22^HA*) and control (*Opn4^Cre; Brn3b^+/+; Rpl22^HA*) littermates. We identified 1360 transcripts that were differentially expressed in ipRGCs upon knockout of BRN3B (Fig. 1A, Supplementary Datasets 1 and 2), with similar numbers of upregulated and downregulated genes, suggesting that BRN3B regulates the expression of a large group of genes in ipRGCs. BRN3B binding[21] was enriched at genes that were downregulated, but not upregulated, upon Brn3bcKO (Supplementary Fig. 4 and Supplementary Dataset 2), suggesting that BRN3B directly binds target genes for activation, while indirectly repressing other genes. By examining previous scRNAseq analyses of available datasets, we also found that differentially expressed genes identified in Brn3bcKO retinas often showed variable expression across ipRGC subtypes (Fig. 1B, C)[3], raising the possibility that BRN3B may be linked to ipRGC subtype identity. Indeed, when we analyzed expression of BRN3B itself in the best-characterized ipRGC subtypes (M1, M2, and M4), we identified graded expression of BRN3B, with M1 ipRGCs expressing the least BRN3B, followed by increasing levels in M2 and then M4 cells (Fig. 1D).

Given this differential expression of BRN3B, we next assessed whether BRN3B regulates the transcriptional programs that define ipRGC subtypes. We curated the top genes that were enriched in ipRGC subtypes with high levels of *Brn3b* expression (M4, M5, and M6 cells, Brn3b^High) or in ipRGC subtypes with low levels of *Brn3b* expression (M1 and M2 cells, Brn3b^Low) using scRNAseq analyses (Fig. 1E). Strikingly, we found that Brn3bcKO retinas showed decreased expression of genes enriched Brn3b^High ipRGCs, suggesting that BRN3B controls the transcriptional identity of M4, M5, and M6 ipRGCs. In addition, we found increased expression of genes enriched in Brn3b^Low ipRGCs, indicating that BRN3B negatively regulates the transcriptional identity of M1 and M2 ipRGCs (Fig. 1E, F). Because groups of transcription factors often work in concert to orchestrate cell identity[22–24], we also examined the full set of 11 transcription factors that exhibited graded expression across subtypes and marked Brn3b^High or Brn3b^Low ipRGCs. Of these, the expression of 8 were dysregulated upon Brn3bcKO (Fig. 1G), suggesting that BRN3B plays a key role in the transcriptional hierarchy that defines ipRGC subtypes.

We further investigated gene regulation by BRN3B across individual ipRGCs using RNAscope. *Chrna6*, a BRN3B target gene encoding a subunit of the nicotinic acetylcholine receptor, is highly expressed in M4, M5, and M6 ipRGCs but shows only modest expression in subsets of M1 and M2/M3 ipRGCs in control retinas (Supplementary Fig. 1D). Knockout of BRN3B led to reduced *Chrna6* expression across all ipRGCs (Fig. 1H–K), consistent with our observations using TRAPseq. Conversely, *Zcchc12*, another BRN3B-regulated gene encoding a transcriptional activator, is expressed primarily in M1 ipRGCs in control retinas (Fig. 1G and Supplementary Fig. 1D). However, following knockout of BRN3B, we observed an increase in the number of ipRGCs expressing *Zcchc12*, including putative M4 ipRGCs (Fig. 1H–K). These findings suggest that BRN3B not only promotes the expression of certain target genes, but it also restricts the expression of other genes to specific ipRGC subtypes, such as M1 cells.

### BRN3B modulates melanopsin expression
Among the genes expressed in ipRGCs, the melanopsin-encoding gene *Opn4* is crucial for conferring intrinsic photosensitivity[9–12,25]. Moreover,

*Opn4* expression varies across ipRGC subtypes, with M1 ipRGCs having the highest melanopsin levels, followed by M2, and then M4 cells, with graded expression levels opposite to that of BRN3B (Fig. 1D)[13,25,26]. The inverse relationship between BRN3B and *Opn4* expression across M1, M2, and M4 ipRGCs prompted the hypothesis that the transcription factor BRN3B regulates *Opn4*'s graded expression patterns across ipRGC subtypes. To test this idea, we quantified *Opn4* expression across individual ipRGCs in both retinal sections and retinal whole mounts of adult Brn3bcKO and control (*Opn4^Cre; Brn3b^+/+*) littermates using mRNA FISH. We observed a significant increase in *Opn4* mRNA expression in M1, M2, and M4 ipRGCs in Brn3bcKO retinal sections and flat mounts (Fig. 2A–C and Supplementary Fig. 5). We observed a similar increase in *Eomes* mRNA expression in Brn3bcKO mice, a known regulator of melanopsin expression[27,28] (Supplementary Fig. 6). Immunohistochemical labeling for melanopsin in Brn3bcKO retinas likewise showed higher melanopsin expression, with increased proportions of ipRGCs expressing high levels of melanopsin (Fig. 2D–F). The total number of total ipRGCs and of individual ipRGC subtypes remained unchanged (Supplementary Fig. 5C and Supplementary Fig. 7C). These results indicate that BRN3B plays an important role in suppressing *Opn4* expression levels in ipRGCs.

In addition to its developmental functions in shaping ipRGC features, BRN3B continues to be expressed in the adult retina, where it may continue to affect gene expression[16]. We therefore tested whether BRN3B regulates *Opn4* expression in ipRGCs beyond development by generating a tamoxifen (TMX)-inducible (i)Brn3bcKO line (*Opn4^CreERT2; Brn3b^cKOAP/cKOAP*) to remove BRN3B from ipRGCs during adulthood. TMX injection over five consecutive days into adult P60 iBrn3bcKO animals resulted in a significant reduction at P120 of *Brn3b* mRNA in ipRGCs compared to animals treated with vehicle control (Supplementary Fig. 8). Strikingly, *Opn4* mRNA levels were significantly increased in ipRGCs of P120 iBrn3bcKO animals compared to those of control littermates (Fig. 2C). Immunohistochemical labeling also showed an increased proportion of ipRGCs expressing high levels of *Opn4* protein upon TMX-induced knockout of BRN3B in adult animals (Fig. 2G, H). These data suggest that BRN3B is required for maintaining ipRGC subtype-specific *Opn4* expression in the adult retina. Collectively, these findings reveal that BRN3B fine-tunes the gene expression profiles of ipRGC subtypes to shape their identity during development and through adulthood.

### BRN3B is a central regulator of morphological features that define ipRGC subtypes
Cellular morphology is a key distinguishing feature of ipRGC subtypes, which we observed to also correlate with BRN3B expression levels. M4 ipRGCs express high levels of *Brn3b* and have large somata and large, complex dendritic arbors (Fig. 1D)[12,29]. M1 ipRGCs, on the other hand, express low levels of *Brn3b* and have small somata and smaller, less complex dendritic arbors. M2 ipRGCs exhibit intermediate characteristics between M4 and M1 cells. Given the key roles of BRN3B in gene regulation associated with ipRGC identity, we next examined whether BRN3B shapes the morphology of ipRGC subtypes. We first compared the morphology of M4 and M2 ipRGCs in Brn3bcKO and control retinas, which both contain the *Opn4^Cre* allele, by intravitreally injecting a diluted, Cre-dependent AAV2-hSyn-DIO-hM3Gq-mCherry virus. This allowed for sparse mCherry labeling of M4 (identified as mCherry-positive, SMI-32-positive) and M2 (identified as mCherry-positive, SMI-32-negative) ipRGCs (Fig. 3A, B and Supplementary Fig. 9)[12,29,30]. M1 cells are not labeled at this low titer[29] (and see "Methods"). Strikingly, M2 and M4 ipRGCs in Brn3bcKO retinas showed a shift toward morphological features reminiscent of M1 cells, including significant reductions in dendritic arbor diameter, complexity, and total length, as well as soma diameter (Fig. 3A, D and F). For M4 cells, these changes were most pronounced in the nasal retina, which is the region in control retinas with the largest and most complex M4 cells

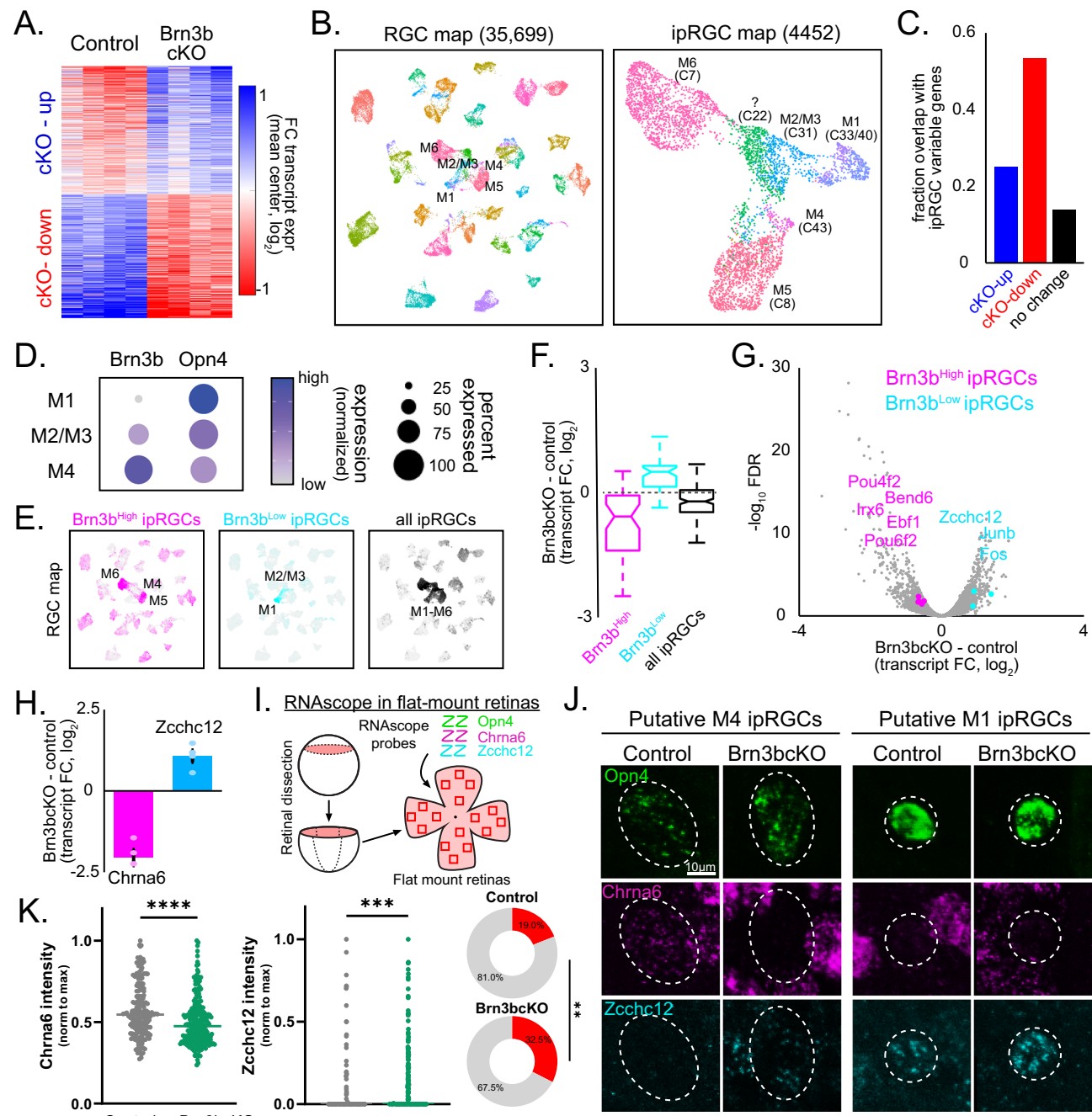

**Fig. 1 | BRN3B confers molecular identity to ipRGC subtypes. A** Differentially expressed transcripts in control and Brn3bcKO ipRGCs. **B** ipRGCs identified from publicly available scRNA-Seq profiles of RGCs[3], re-analyzed using dimensionality reduction, and visualized with UMAP. Established genetic markers are used to annotate ipRGC subtypes. **C** Genes with increased and decreased expression in Brn3bcKO mice exhibited high variability in expression across ipRGC subtypes. **D** *Brn3b* and *Opn4* mRNA levels in ipRGC subtypes. Relative expression of *Opn4* and *Brn3b* mRNA in M1, M2, and M4 ipRGCs in scRNA-Seq dataset[3]. **E** Top genes that were enriched in M4, M5, and M6 ipRGCs with high levels of *Brn3b* expression (Brn3b^High) or in M1 and M2 ipRGC subtypes with low levels of *Brn3b* expression (Brn3b^Low)[3]. **F** Brn3bcKO induces opposite effects on transcripts found in Brn3b^Low and Brn3b^High ipRGCs (*n* = 50,57,164 transcripts for Brn3b^High, Brn3b^Low, all ipRGCs). Box plots show median, quartiles (box) and range (whiskers). **G** Volcano plot of

Brn3bcKO-induced changes in transcript expression using TRAP. Transcriptional regulators which define ipRGC subtypes and are dysregulated in Brn3bcKO are highlighted. (H) Brn3bcKO-induced changes in *Chrna6* and *Zcchc12* transcript expression using TRAP (*n* = 4 mice/group). Data show mean ± standard error. **I** Schematic representation of RNAscope in flat-mount retinas. **J** Representative RNAscope pictures in flat mount retinas showing *Chrna6* and *Zcchc12* mRNA expression in putative M4 and M1 ipRGCs (dashed ellipses) in control (*Opn4^Cre/+*; *Brn3b^+/+*) and Brn3bcKO mice. **K** Relative expression of *Chrna6* (*P* = 0.0001) and *Zcchc12* (*P* = 0.0002) mRNA in ipRGCs from control (gray, *n* = 183 cells) and Brn3bcKO (green, *n* = 242 cells) mice measured by RNAscope. Pie plots show an increased number of ipRGCs expressing *Zcchc12* mRNA in Brn3bcKO mice (*P* = 0.003). Source data are provided as a Source Data file. Lines are median values, **P* < 0.01, ***P* < 0.001, two-tailed Mann-Whitney U and Fisher's exact tests.

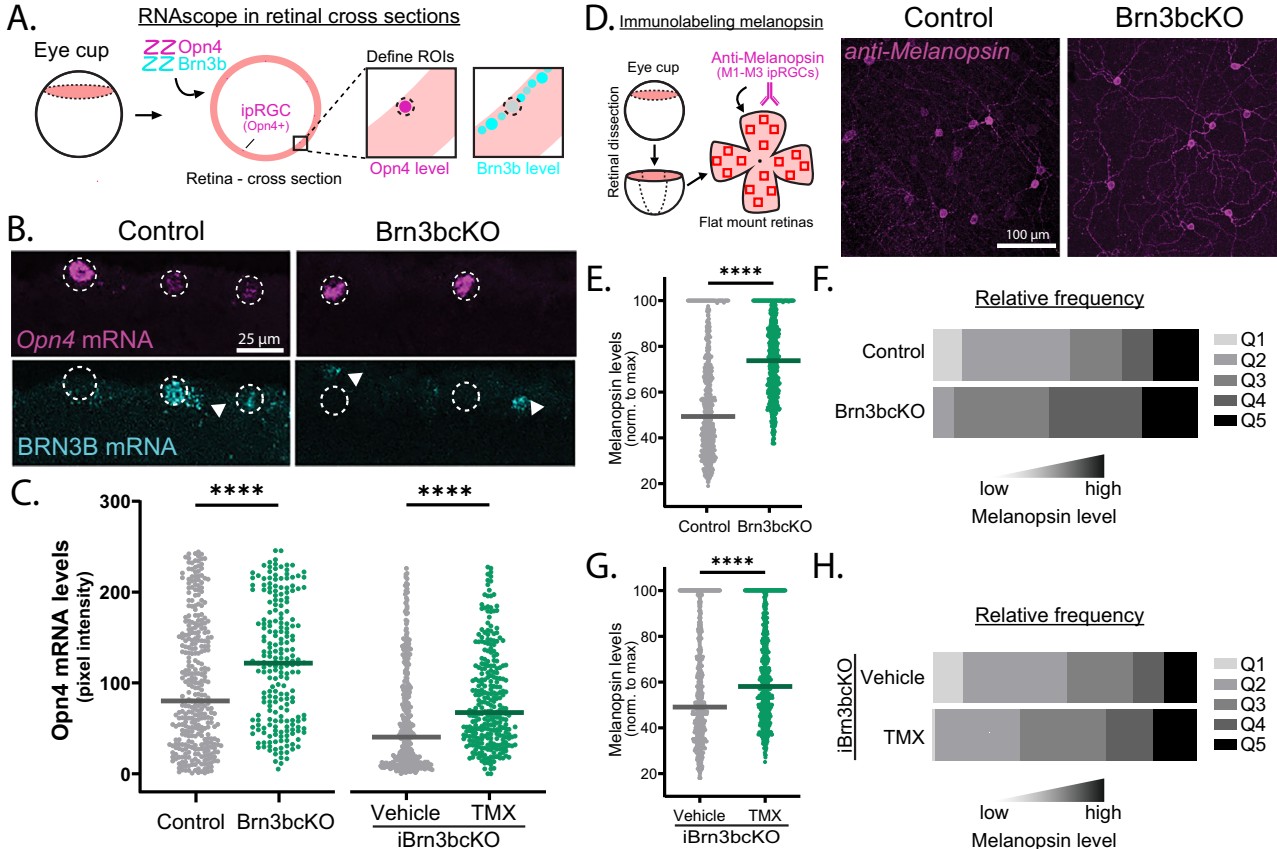

**Fig. 2 | BRN3B represses melanopsin expression. A** Schematic representation of the RNAscope retinal cross-sections protocol. **B** *Opn4* and *Brn3b* mRNA expression in control (*Opn4*<sup>Cre/+</sup>; *Brn3b*<sup>+/+</sup>) and Brn3bcKO retinal sections. *Brn3b* mRNA expression is absent in Brn3bcKO ipRGCs (dashed ellipse) and still present in other RGCs (arrowheads). **C** (Left) *Opn4* mRNA levels are significantly increased in ipRGCs Brn3bcKO (green, *n* = 216 cells) compared to control (black, *n* = 336 cells) littermates (*P* = 0.0001). (Right) *Opn4* mRNA levels are increased in iBrn3bcKO (green, *n* = 288 cells) compared to control ipRGCs (black, *n* = 371 cells) (*P* = 0.0001). **D** (Left) Schematic representation of melanopsin immunolabeling in flat-mount retinas. (Right) melanopsin immunolabeling in ipRGCs of adult control and

Brn3bcKO retinas. **E**, **F** melanopsin levels are increased in Brn3bcKO ipRGCs (**E**) (*n* = 600 cells/group, *P* = 0.0001), with an increased proportion of ipRGCs expressing high levels of melanopsin (**F**) (*n* = 600 cells/group). Q1–Q5: quintiles; showing low (Q1) to high (Q5) melanopsin levels. **G**, **H** melanopsin levels are increased in iBrn3bcKO compared to control ipRGCs (**G**) (*n* = 800 cells in Vehicle; *n* = 761 in TMX, *P* = 0.0001), with an increased proportion of ipRGCs expressing high levels of melanopsin (**H**) (*n* = 800 cells in Vehicle; *n* = 761 in TMX). Source data are provided as a Source Data file. Lines are median values, ****P* < 0.001, two-tailed Mann Whitney U test.

(Supplementary Fig. 9[29]). Notably, nearly all labeled M2 and M4 cells in Brn3bcKO retinas remain stratified in the ON sublamina (~10% stratified in the OFF sublamina, Supplementary Fig. 10), indicating that BRN3B is not required for lamination of M2 and M4 ipRGCs. These data suggest that BRN3B establishes the dendritic and somatic morphology of M2 and M4 ipRGCs that distinguishes them from M1 ipRGCs.

M1 ipRGCs also express *Brn3b*, albeit at much lower levels than M4 or M2 cells (Fig. 1D)[16]. Indeed, the suprachiasmatic nucleus-projecting M1 (SCN-M1) ipRGCs reportedly express little to no *Brn3b*[16], which led us to predict that SCN-M1 morphology may be minimally impacted or unaffected in Brn3bcKO animals. To test this, we compared the cellular morphology of SCN-M1 ipRGCs in Brn3bcKO versus control retinas. We labeled SCN-M1 ipRGCs using a retrograde, Cre-dependent AAV (rgAAV-hSyn-DIO-hM3Gq-mCherry) injected into the SCN to visualize the dendritic arbor and soma of SCN-M1 cells (Fig. 3C). Contrary to our expectations, we found that SCN-M1 ipRGCs in Brn3bcKO animals have significantly smaller dendritic arbor size, total length, and complexity, as well as significantly smaller soma diameter than control littermates (Fig. 3C–F). These differences were most pronounced in SCN-M1 ipRGCs of the ventral retina (Supplementary Fig. 11).

These BRN3B-dependent changes in SCN-M1 morphology led us to examine whether SCN-M1 ipRGCs might in fact express BRN3B. We tested this by performing mRNA FISH labeling of BRN3B mRNA in

retinas of *Opn4*<sup>Cre</sup> animals, where SCN-M1 ipRGCs were retrogradely labeled as described above. Consistent with the observed morphological changes, we detected low levels of BRN3B expression in SCN-M1 ipRGCs (Supplementary Fig. 12). Notably, ventral SCN-M1 cells showed significantly higher BRN3B mRNA labeling than their dorsal counterparts, supporting our observation that ventral SCN-M1 cells of the Brn3bcKO retinas showed more drastic changes in morphology compared to those in the dorsal retina (Supplementary Fig. 12). These findings demonstrate that SCN-M1 ipRGCs express modest mRNA levels of *Brn3b*, and that even at these lower levels, BRN3B is crucial for regulating both dendritic and somatic morphology. Collectively, our data show that BRN3B expression drives the development of larger somata and larger, more complex dendritic arbors to shape the morphological features that define ipRGC subtype identity.

## BRN3B defines physiological properties of M4 ipRGCs

Intrinsic physiological properties vary across ipRGC subtypes and shape how features of the visual scene are relayed to downstream areas in the brain. Intriguingly, two key properties, input resistance (M4 < M2 < M1), resting membrane potential (M4 ≈ M2 < M1) and maximum evoked spike frequency (M4 > M2 > M1) correlate with BRN3B-expression levels across M1, M2, and M4 ipRGCs (Supplementary Fig. 13)[12,25]. We therefore tested whether loss of BRN3B would alter

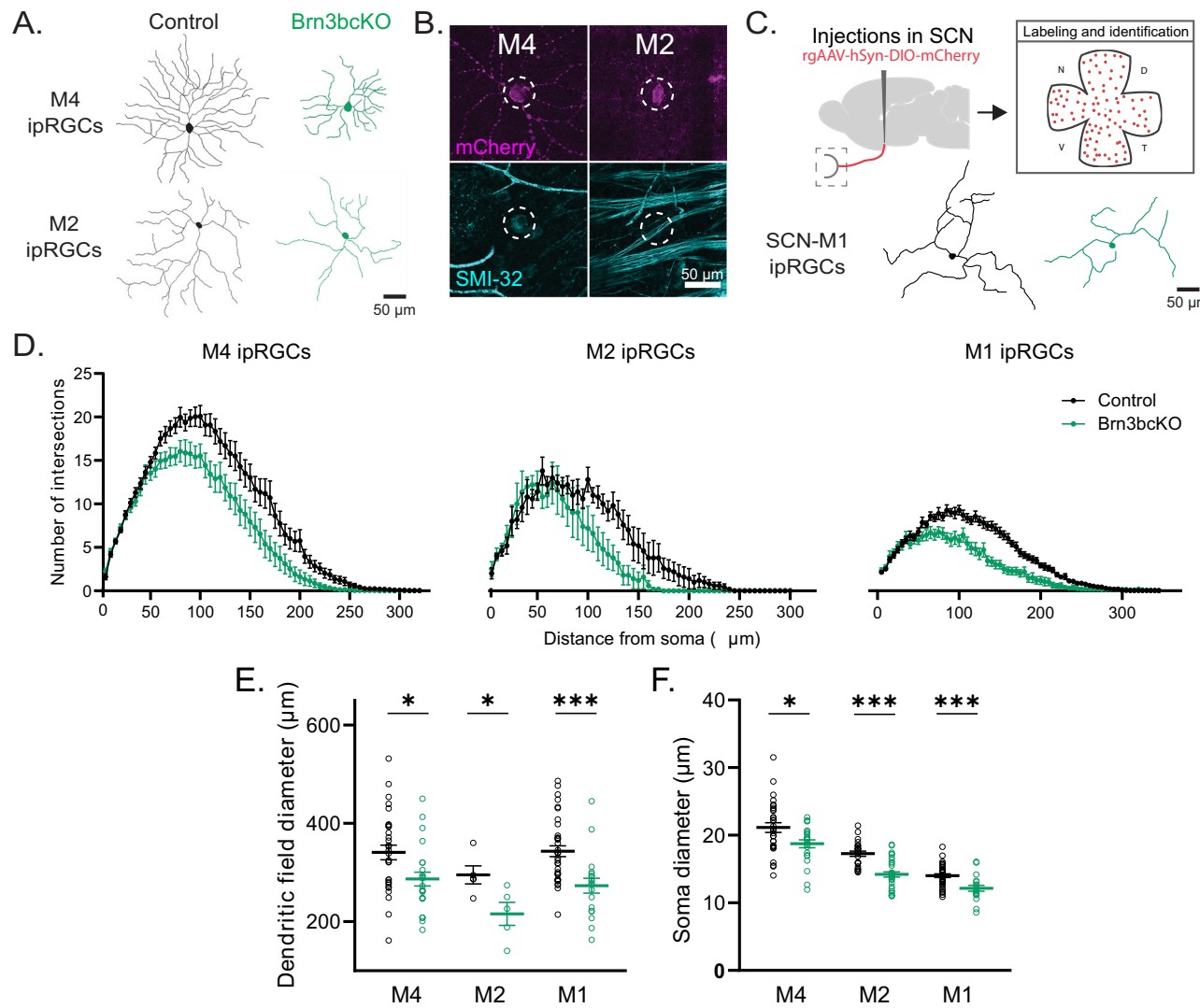

**Fig. 3 | BRN3B tunes morphological properties of ipRGC subtypes.**
**A** Representative dendritic arbor tracing of M4 and M2 ipRGCs in control (*Opn4*$^{Cre/+}$; *Brn3b*$^{+/+}$) and Brn3bcKO mice. **B** Subtype identified by presence (M4) or absence (M2) of SMI-32 immunolabeling (dashed ellipses), examples are from Brn3bcKO cells. **C** (Top) Scheme of retrogradely labeling strategy of M1 ipRGCs. (Bottom) Representative dendritic arbor tracing of M1 ipRGCs in control and Brn3bcKO mice. **D** Sholl analysis of M4 (*n* = 30 cells in Control, *n* = 23 cells in Brn3bcKO), M2 (*n* = 5 cells per group) and M1 (*n* = 36 cells in Control, *n* = 18 cells in Brn3bcKO) ipRGC subtypes. ipRGCs from Brn3bcKO (green) mice present less complex dendritic arbors compared to control (black) mice. **E**, **F** Brn3bcKO showed decreased dendritic field diameter (**E**) (*n* = 30 cells in Control M4, *n* = 23 cells in Brn3bcKO M4, *P* = 0.013; *n* = 5 cells in M2 Control and Brn3bcKO, *P* = 0.028; and Brn3bcKO, *n* = 36 cells in Control M1 and *n* = 18 cells in Brn3bcKO M1, *P* = 0.0004) and soma size (**F**) (*n* = 30 cells in Control M4, *n* = 23 cells in Brn3bcKO M4, *P* = 0.016; *n* = 24 cells in Control M2, *n* = 32 cells in Brn3bcKO, *P* = 0.0001; and Brn3bcKO, *n* = 36 cells in Control M1 and *n* = 19 cells in Brn3bcKO M1, *P* = 0.0004) in M4, M2 and M1 ipRGC subtypes than control mice. Source data are provided as a Source Data file. All data are Mean ± SEM, \**P* < 0.05, \*\*\**P* < 0.001, two-tailed Student's t- and Mann Whitney U tests.

these properties in M1, M2, and M4 ipRGCs. We performed whole-cell patch clamp recordings of M1, M2, and M4 ipRGCs in Brn3bcKO and control littermates in the presence of synaptic blockers to isolate the intrinsic physiological properties of the cells (Fig. 4A and Supplementary Fig. 14). M4 ipRGCs showed significantly increased input resistance in Brn3bcKO retinas (Fig. 4B and Supplementary Fig. 15) and increased resting membrane potential in M2 ipRGCs (Fig. 4C). Brn3bcKO M4 cells also reached significantly lower firing rates when injected with positive current compared to those of littermate controls (Fig. 4D and G and Supplementary Fig. 16). One notable feature of M4 ipRGC firing that further distinguishes it from that of M1 and M2 cells is that even large injections of positive current fail to drive M4 cells into depolarization block (Fig. 4D–I and Supplementary Fig. 16)[12,25,30]. However, Brn3bcKO M4 cells did in fact reach depolarization block when injected with positive current, indicating a shift in M4 cell

properties toward those of M1 and M2 cells (Fig. 4G and Supplementary Fig. 16). These differences were still detectable even when normalizing for current density to account for cell size differences, indicating that the spiking properties of M4 cells are affected by loss of BRN3B, and that this phenotype is not secondary to a decrease in cell size (Supplementary Fig. 16). Similar trends were observed in M2 and M1 cell spiking patterns (Fig. 4).

Interestingly, the melanopsin photocurrent of Brnb3cKO M2 ipRGCs showed faster kinetics at light response onset, though the amplitude of the intrinsic photocurrent was unchanged overall. The intrinsic photocurrent in M4 and M1 cells was unchanged in amplitude or kinetics, indicating that changes in melanopsin and BRN3B expression in Brn3bcKO animals do not result in changes to the saturated photocurrent in ipRGC subtypes (Supplementary Fig. 17 and Supplementary Fig. 18). Overall, our data show that BRN3B regulates

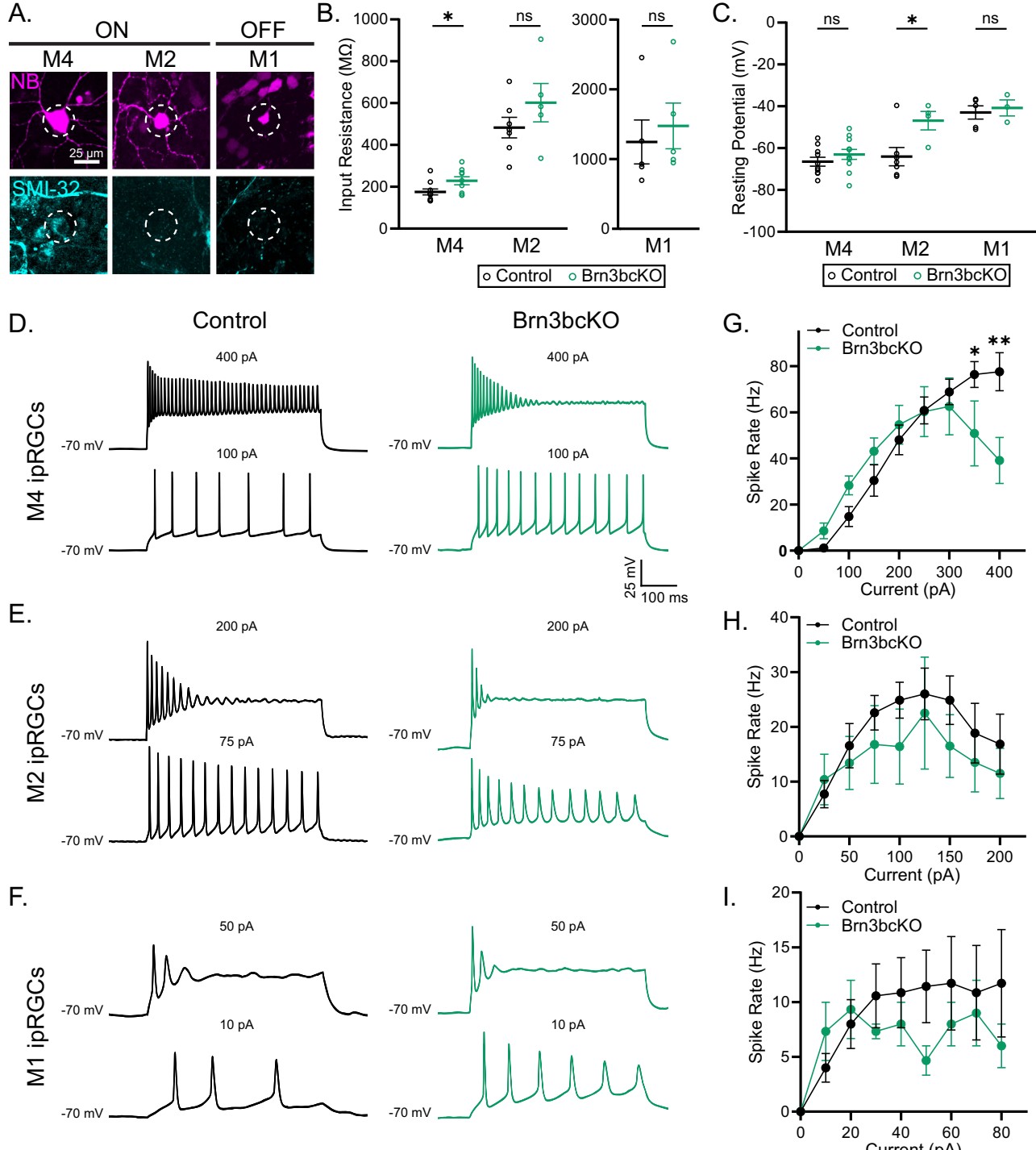

**Fig. 4 | BRN3B tunes physiological properties of ipRGC subtypes.**
**A** Representative pictures of M4, M2 and M1 ipRGCs (dashed ellipses) filled with neurobiotin (NB) used for electrophysiological recordings. **B** Brn3bcKO M4 ipRGCs (green, $n = 9$ cells) showed increased input resistance compared to control M4 ipRGCs (black, $n = 10$ cells) ($P = 0.04$), while no differences were observed in M2 ($n = 7$ cells in Control, $n = 5$ cells in Brn3bcKO; $P = 0.24$) and M1 ($n = 7$ cells in Control, $n = 5$ cells in Brn3bcKO; $P = 0.45$) ipRGCs. **C** M2 ipRGC subtype in Brn3bcKO retinas showed increased resting membrane potential ($n = 7$ cells in Control, $n = 4$ cells in Brn3bcKO; $P = 0.03$) while no differences were observed in M4 ($n = 10$ cells in Control, $n = 11$ cells in Brn3bcKO; $P = 0.29$) and M1 ($n = 7$ cells in Control, $n = 3$

cells in Brn3bcKO; $P = 0.69$) ipRGCs. **D–F** representative excitability traces from M4 (**D**), M2 (**E**) and M1 (**F**) ipRGCs in Control (black) and Brn3bcKO (green) retinas. **G** Brn3bcKO M4 ipRGCs reached peak firing rate by 300pA and then showed marked depolarization block ($n = 5$ cells in Control, $n = 7$ cells in Brn3bcKO; 350 pA $P = 0.02$, 400 pA $P = 0.001$). **H, I** No differences were observed in excitability experiments in M2 (**H**, $n = 7$ per group) and M1 (**I**, $n = 7$ in Control, $n = 3$ in Brn3cKO) ipRGCs. Source data are provided as a Source Data file. All data are Mean ± SEM, *$P < 0.05$, ***$P < 0.001$, two-tailed Student's t- and Mann Whitney U tests and two-way repeated measures ANOVA with Tukey's multiple comparisons test.

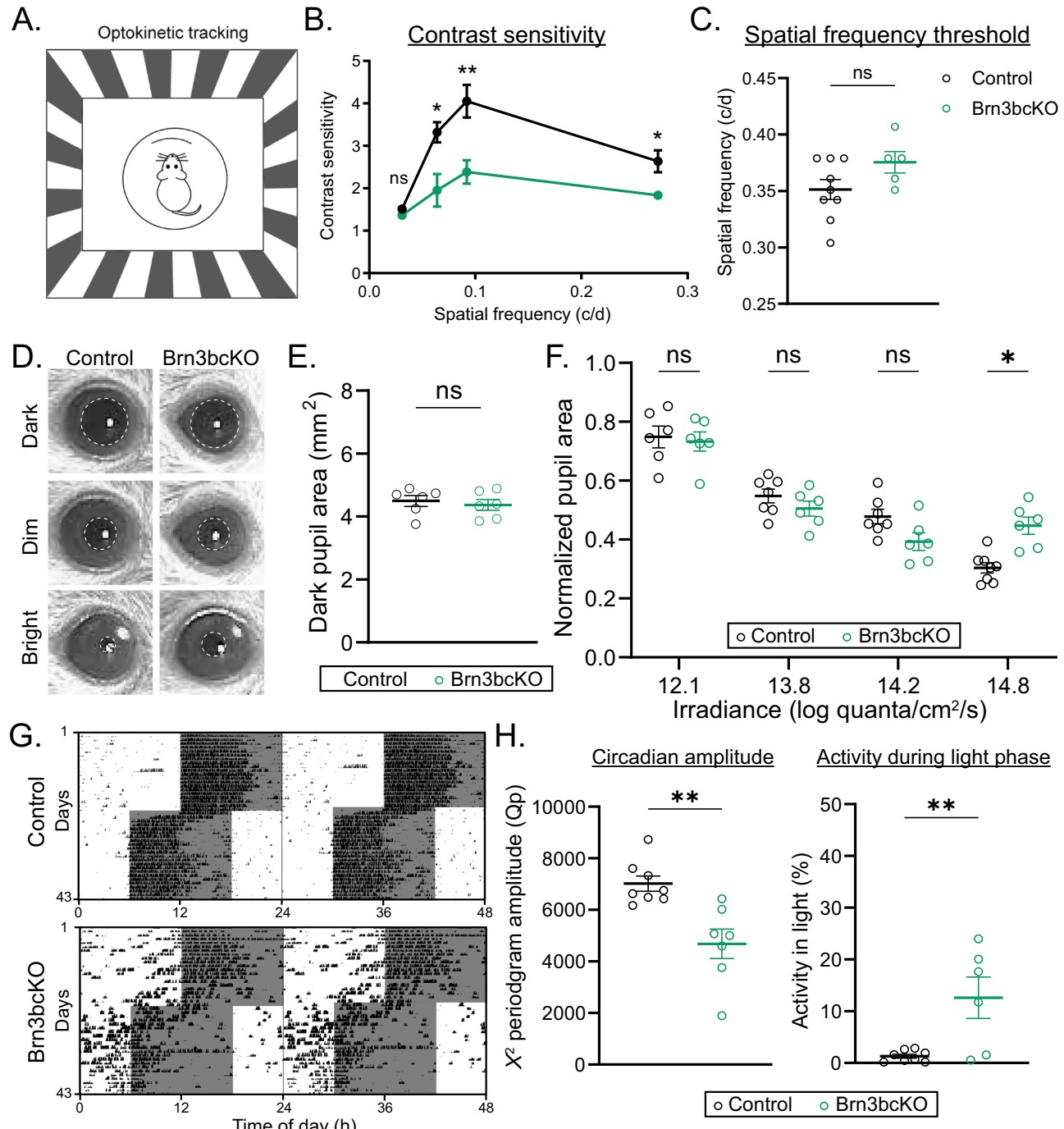

**Fig. 5 | BRN3B is a central regulator of ipRGC-driven behaviors. A** Schematic showing the optokinetic tracking (OKT) behavioral assay. **B** Contrast sensitivity curve at different spatial frequencies. Brn3bcKO mice (green, $n = 6$) showed decreased contrast sensitivity compared to control ($Opn4^{Cre/+}$; $Brn3b^{+/+}$) mice (black, $n = 12$) (0.031 cpd, $P = 0.176$; 0.064 cpd, $P = 0.006$; 0.092 cpd, $P = 0.011$; 0.272, $P = 0.128$). **C** Spatial frequency threshold measured by OKT at 100% contrast in control (black, $n = 9$) and Brn3bcKO (green, $n = 5$) mice. No differences between Brn3bcKO and control littermates ($P = 0.106$). **D** Representative PLR images from control (left) and Brn3bcKO (right) mice in darkness (top), dim light (middle, 13.8 log quanta cm$-2$ s $-1$), and bright light (bottom, 14.8 log quanta cm$-2$ s $-1$). Pupils are highlighted with dashed ellipses. **E** Control and Brn3bcKO ($n = 6$ mice/group) pupil area in the dark ($P = 0.626$). **F** Irradiance-response relationship of PLR in control (black) and Brn3bcKO (green) mice ($n = 6$ mice/group) (12.1 log quanta/cm$^2$/s, $P = 0.997$; 13.8 log quanta/cm$^2$/s, $P = 0.673$; 14.2 log quanta/cm$^2$/s, $P = 0.190$; 14.8 log quanta/cm$^2$/s, $P = 0.010$). **G** Representative double-plotted actograms from control (top) and Brn3bcKO (bottom) mice. Mice were initially exposed to a 12:12 LD cycle with 100 lux light during the light phase. Then, the mice were exposed to a 6 h phase advance. **H** Circadian amplitude measured using the peak amplitude of the $\chi^2$ periodogram in control (black, $n = 8$) and Brn3bcKO mice (green, $n = 7$) ($P = 0.002$). (Left); and percent activity during the light phase (Right) in control (black, $n = 8$) and Brn3bcKO mice (green, $n = 7$) ($P = 0.006$). Source data are provided as a Source Data file. All data are Mean ± SEM, n.s. (not significant) $P > 0.05$; *$P < 0.05$; **$P < 0.01$. Two-tailed Mann-Whitney U test and two-way repeated measures ANOVA with Tukey's multiple comparisons test.

both excitability and input resistance of ipRGC subtypes and alters kinetics of the photocurrent in M2 ipRGCs, all key subtype-defining features of ipRGCs.

### ipRGC-dependent behaviors are altered in Brn3bcKO animals

The transcriptional and morphophysiological features that define ipRGC subtypes are key to their role in diverse visual behaviors, including contrast sensitivity, pupillary light reflex (PLR), and circadian photoentrainment. Given our findings on the critical role of BRN3B in ipRGC subtype transcriptional identity, morphology, and intrinsic physiological properties, we next examined whether ipRGC-dependent behaviors would be altered in Brn3bcKO animals. Contrast sensitivity is crucial for pattern vision and visual perception, and disruption or ablation of non-M1 ipRGCs reduces contrast sensitivity thresholds[12]. Using the Optomotry system, we found that Brn3bcKO animals showed significantly decreased contrast sensitivity compared to control littermates, but no change in spatial frequency threshold, indicating that normal contrast sensitivity depends on BRN3B expression in ipRGCs (Fig. 5A–C and Supplementary Fig. 19).

Pupil constriction is an M1 ipRGC-dependent behavior that regulates the amount of light entering the eye and improves visual function, allowing animals to see across a broad range of environmental light intensities[11,16,31]. We therefore next tested whether the pupillary light reflex was altered in Brn3bcKO animals by comparing consensual pupil constriction Brn3bcKO and control mice across a range of light intensities at 470 nm (12.1–14.8 log quanta/cm²/s) (Fig. 5D–F). Brn3bcKO animals showed significantly less pupil constriction at the brightest light intensity but no change in kinetics or dark pupil diameter, suggesting that BRN3B-dependent regulation of M1 ipRGC features may be required for normal pupil constriction in bright light (Fig. 5F and Supplementary Fig. 20).

Another behavior regulated by M1 ipRGCs is circadian photoentrainment, the synchronization of an organism's body clocks to environmental light/dark cycles to properly time activity and physiology rhythms. Misalignment of these rhythms with environmental light/dark cycles has severe consequences for mental and physical health and animal survival[32,33]. We assessed whether proper photoentrainment is dependent on BRN3B expression in ipRGCs by tracking voluntary wheel-running activity of littermate Brn3bcKO and control littermates in a 12 h:12 h light:dark (LD) cycle at 100 lux. All mice photoentrained with a similar period length and total daily activity (Supplementary Fig. 16). However, we observed significantly lower circadian amplitudes in Brn3bcKO mice, indicating that loss of BRN3B in ipRGCs dampens photoentrainment (Fig. 5G, H and Supplementary Figs. 21, 22). In addition, Brn3bcKO mice showed significantly higher activity during the light phase of the LD cycle and more variable, phase-advanced activity onset, suggesting impaired light signaling to the SCN (Fig. 5H and Supplementary Fig. 21). This data indicates that BRN3B expression in SCN-M1 ipRGCs is required for their proper function in circadian photoentrainment. Together, our findings unveil the critical roles of BRN3B in the genetic and morphophysiological tuning of ipRGC subtypes, that is critical in shaping their proper function across multiple visual behaviors.

## Discussion

A unique blend of molecular and cellular features enables each ipRGC subtype to precisely regulate downstream brain circuits and behaviors that operate over a range of timescales and environmental contexts. The transcriptional programs underpinning this neuronal diversity were previously unknown. Here, we identify graded expression of the transcription factor BRN3B across ipRGC subtypes as a key mechanism in tuning their genetic and morphophysiological features. Loss of this pattern causes the distinguishing features of ipRGCs to shift toward those of M1 cells. Higher levels of BRN3B expression in M4 ipRGCs differentiate these cells from M1 and M2 ipRGCs, and knockout of

BRN3B shifts the properties of M4 cells towards those of M1 and M2 cells. Notably, it is still possible to distinguish ipRGC subtypes in the Brn3bcKO line, and subtypes appear in similar numbers, indicating that, though properties are altered, subtype identity is not fully transformed. The cellular changes in ipRGCs following BRN3B knockout result in deficits in several visual behaviors, including in contrast sensitivity, a behavior linked to M4 ipRGCs[16]. While the RGC scRNAseq atlas suggests some Brn3b expression levels in M1 ipRGCs, mouse genetic studies have suggested that SCN-M1 ipRGCs are largely Brn3b-negative. We find that even SCN-M1 ipRGCs show low levels of BRN3B expression that are critical for shaping cellular properties and their role in circadian photoentrainment, highlighting its critical role in defining the full spectrum of ipRGC subtypes. SCN-M1 cells are spared in mouse lines where diphtheria toxin expression from the BRN3B promoter is driven in ipRGCs, which had resulted in the conclusion that these cells did not express BRN3B. However, our data suggest that SCN-M1 survival in this line is simply due to low expression of BRN3B and therefore diphtheria toxin and that even this low BRN3B expression shapes the features and resulting behavioral functions of SCN-M1 cells. It seems plausible that the increased propensity for depolarization block in all ipRGC subtypes, along with the increased melanopsin expression (and presumably sensitivity to light), could together compress the dynamic range of ipRGC signaling, impacting downstream behaviors. It is also notable that these shifts in ipRGC subtype features are visible despite imperfect excision of the *Brn3b* gene, suggesting that the phenotypes are an underrepresentation of the full scope of BRN3B impact. Importantly, loss of BRN3B does not appear to reduce ipRGC number or the total number of ipRGC subtypes (Supplementary Figs. 5C, I and 7C) when we directly label for melanopsin expression, though HA labeling suggests a slight decrease in the Ribotag line. Given that this is a more indirect measure involving multiple recombination steps, we tend to weigh these results less heavily, though they are important to note.

Our findings also suggest that ipRGC subtype identity is shaped by multiple transcription factors, many of which are under the control of BRN3B. Notably, BRN3B fine-tunes many ipRGC features, including melanopsin expression and morphophysiological properties, but it's removal does not result in homogeneity of these defining features, suggesting that although BRN3B has broad effects, additional regulators also shape these subtype-defining ipRGC features. It will be interesting to determine how ipRGC features that develop independently of BRN3B activity, including the laminar positioning of ipRGCs, are genetically regulated. Intriguingly, BRN3B continues to modify ipRGC identity from early development through adulthood, partly by fine-tuning melanopsin expression. Whether extrinsic cellular signals and experiences in animals may control BRN3B activity to regulate the cellular functions of ipRGCs even into adulthood remains to be elucidated. While the majority of canonical RGCs do not express BRN3B postmitotically, a subset do continue to express it. It will be important to determine how melanopsin suppression occurs and morphophysiological features are shaped in a BRN3B-dependent or independent manner in specific canonical RGC types. Together, our study opens new windows into the genetic tuning of retinal cell types and paves the way for future studies on how molecular programs functionally shape cellular identity in the brain.

## Methods

### Animals

All procedures were approved by the Animal Care and Use Committee at Northwestern University (Protocol number: IS00003845). Animals were housed in a vivarium under 12:12 light/dark cycle conditions with *ad libitum* access to food and water. The temperature ranged from 21 to 23 °C, and the humidity ranged from 30% to 70%. Male and female mice were used with a mixed B6/129 background for all experiments at embryonic (E13.5-E15) and adult (P40-P150) stages. For RNAscope

Fluorescence in situ hybridization and immunohistochemical experiments, we utilized *WT*, *Opn4*[Cre/+13] (RRID:IMSR_JAX:035925), *Opn4*[Cre/+]; *Brn3b*[cKOAP/cKOAP] [15], (RRID: IMSR_JAX:010559) and *Opn4*[CreERT2/+], *Brn3b*[cKOAP/cKOAP] mice[16], (RRID: IMSR_JAX:035926). For morphological and physiological experiments, we used *Opn4*[Cre/+], *Opn4*[Cre/+]; *R26R-EYFP* (RRID:IMSR_JAX:006148) and *Opn4*[Cre/+]; *Brn3b*[cKOAP/cKOAP] mice. TRAPseq experiments were conducted using *Opn4*[Cre/+] *Rpl22*[HA18], (RRID: IMSR_JAX:029977) and *Opn4*[Cre/+]; *Brn3b*[cKOAP/cKOAP] *Rpl22*[HA] mice. For behavioral experiments, we utilized *Opn4*[Cre/+] and *Opn4*[Cre/+]; *Brn3b*[cKOAP/cKOAP] mice. Mice were anesthetized with either Avertin or isoflurane, depending on the procedure. Cervical dislocation was used as a secondary method of euthanasia following deep anesthesia. Post-surgical animals were monitored daily for at least 72 h to assess health, behavior, and signs of discomfort or distress. All efforts were made to minimize pain and discomfort throughout the experiments.

## TRAP-seq

Eyes from postnatal day 60 (P60) mice expressing HA-Rpl22 in ipRGCs or control mice were enucleated, and eyecups were dissected in cold nuclease-free 1X PBS with 100 µg/ml cycloheximide. Retinas from the same animal were combined and homogenized in lysis buffer (50 mM Tris-HCl, pH 7.4, 100 mM KCl, 12 mM MgCl$_2$, 1% NP-40, 0.4 unit/ml RNase inhibitor (Promega, N2511), 1 mM DTT, 100 µg/ml cycloheximide) with a dounce homogenizer. Lysates were incubated with 1 µg/µl heparin (Sigma, H3393) for 2 min on ice prior to centrifugation. The lysates were spun down at 10,000 × g for 10 min at 4 °C and the supernatant was used for immunoprecipitation using an antibody against HA-tag (Sigma, H3663) with Dynabeads protein G (Invitrogen) for overnight at 4 °C. 1.5% of the lysate was saved as an input. After extensive washes with cold washing buffer (50 mM Tris-HCl pH 7.4, 300 mM KCl, 12 mM MgCl$_2$, 1% NP-40, 0.5 mM DTT, 100 µg/ml cycloheximide), immunoprecipitated RNA and input RNA were purified with a RNeasy Micro Kit (Qiagen) following the manufacturer's instructions and eluted in 10 µl of water. 2 µl of purified RNA were used for sequencing library preparation using the Smart-seq3 protocol[34] with modifications. After reverse transcription followed by PCR amplification, 25 ng of cDNA was used for the Tn5 tagmentation reaction. Tagmented cDNA was amplified with 8 cycles of PCR using primers containing indexes and Illumina adapters. The RNA libraries were quantified by Qubit (Thermo Fischer Scientific) and bioanalyzer (Agilent) and sequenced on the Illumina NextSeq 550 platform to obtain 37 bp paired-end reads. Two to four biological replicates were performed for all conditions.

## RNA-seq analyses

For TRAP-seq analyses, Smart-seq3 prepared libraries were aligned to the mm10 reference genome using the Smart-seq3 github pipeline (https://github.com/sandberg-lab/Smart-seq3)[34]. Transcripts considered for analyses were selected based on higher normalized counts following HA immunoprecipitation in HA-*Rpl22* expressing retina compared to HA immunoprecipitation in control retina. Differential gene expression analyses were performed using pair-wise negative binomial tests with edgeR (RRID:SCR_012802)[35], and the false discovery rate (FDR) was calculated for all genes.

The top 100 genes enriched or de-enriched in ipRGCs following HA-*Rpl22* immunoprecipitation compared to total retinal RNA were used to generate modules for visualization on the scRNA-seq UMAP plot using Seurat 4 (RRID:SCR_016341)[36]. Genes upregulated or downregulated in Brn3bcKO ipRGCs compared to control ipRGCs (log$_2$ fold change > 1 or <−1) or not changing (absolute log$_2$ fold change < 0.05) following HA-*Rpl22* immunoprecipitation were intersected with the top 2000 genes with the highest standardized variance in ipRGCs calculated using Seurat 4[36]. The fraction overlap was derived using the number of overlapping genes divided by the number of Brn3bcKO-upregulated, downregulated, or not changing genes. For

comparisons of selected transcripts using HA-*Rpl22* immunoprecipitation of Brn3bcKO and control retinas, we considered the dominant HA-labeled cell types, including ipRGCs and a small group of cone cells. To adjust for changes in the ratio of these cells upon knockout of BRN3B, the transcript FC of ipRGC-enriched genes was normalized by the FC in the number of HA-labeled ipRGCs relative to the number of HA-labeled cones.

BRN3B CUT&Tag datasets from RGCs in E14.5 mice with annotated BRN3B peaks were obtained[21]. The fraction of Brn3bcKO downregulated, upregulated, or not changing genes that harbored BRN3B peaks located within 10Kb of their gene bodies was calculated.

scRNA-seq datasets from RGCs in adult mice with cell type and subtype annotations were obtained[3] and analyzed using Seurat 4[36]. Briefly, all RGCs or ipRGCs were extracted for downstream analyses. Read counts were normalized and integrated across three biological replicates using fastMNN followed by UMAP embedding[37]. As previously described, ipRGC subtype and cluster annotations were M1 (C33/C40), M2 (C31), M4 (C43), and an unclassified subtype (C22)[3]. Moreover, C7 and C8 were included given their similarity in gene expression profiles to the M1-M4 ipRGC subtypes, as also noted in recent studies[38,39]. We denote these as M6 (C7), given its enrichment for *Cdh3*[40] and provisionally as M5 (C8), given its enrichment for a combination of PixON transcripts, including *Gnas* derived from Patchseq analyses[8,41]. Gene module scores were calculated from the average expression of genes within the module, subtracted by the levels of similarly expressed control genes[36,42]. To identify genes expressed in a graded pattern across M1 to M6 ipRGCs, differentially expressed genes in clusters containing M1 and M2/3 ipRGCs were compared with those in clusters containing M4, M5, and M6 ipRGCs and an unidentified group (cluster 22) using the Wilcoxon rank sum test. Genes enriched in M4–M6 ipRGCs and cluster 22, including *Brn3b*, were denoted as Brn3b[High]. Genes enriched in M1 and M2/M3 ipRGCs were denoted as Brn3b[Low].

## Viral infection

For morphological studies, we used two strategies to label different ipRGC subtypes. To sparsely label M4 and M2 ipRGCs[29], *Opn4*[Cre/+] and *Opn4*[Cre/+]; *Brn3b*[cKOAP] mice between P40-60 were anesthetized by intraperitoneal (IP) injection of Avertin, and a 30-gauge needle was used to open a hole in the *ora serrata*. Each eye was intravitreally injected with 1 µL of AAV2-hSyn-DIO-hM3Gq-mCherry (~8 × 10¹¹ GC/mL, Addgene) using a custom Hamilton syringe (Borghuis Instruments) with a 33-gauge needle (Hamilton). To label M1 ipRGCs, mice were anesthetized with isoflurane (Kent Scientific VetFlo system) and then bilaterally injected with 150 nl of rgAAV-hSyn-DIO-hM3Gq-mCherry (~8 × 10¹² GC/mL, cat. #: 44361-AAVrg, Addgene) in the suprachiasmatic nucleus (SCN) (AP: − 0.2, ML: ± 0.15, DV: 5.85) using a stereotaxic injector (Neurostar) controlled by the software Stereodrive at an injection rate of 30 nl/min.

To label ipRGCs for electrophysiological studies *Opn4*[Cre/+] and *Opn4*[Cre/+]; *Brn3b*[cKOAP] mice between P40-60 were anesthetized as described above. Each eye was intravitreally injected with 1 µL of AAV2-hSyn-DIO-mCherry (~8 × 10¹² GC/mL, cat. #: 50459-AAV2, Addgene). After the surgery procedures, mice were subcutaneously injected with 2 mg/kg of meloxicam (Covetrus).

## Tamoxifen injection

Tamoxifen (TMX, Sigma-Aldrich) was dissolved in corn oil at a concentration of 20 mg/ml by shaking overnight at 37 °C, and then the solution was stored at 4 °C for the duration of injections. *Opn4*[CreERT2/+]; *Brn3b*[cKOAP] mice at P60 were IP injected once every 24 hours for a total of 5 consecutive days with 100 µl of tamoxifen or Vehicle (corn oil)[16]. Mice were closely monitored, and no adverse reactions to the treatment were noticed. Mice were euthanized 60 days after the first injection for RNA fluorescence in situ hybridization and immunohistochemical studies.

## RNA fluorescence in situ hybridization (FISH)

Mice were anesthetized by IP injection of Avertin and sacrificed by cervical dislocation. Eyecups were dissected in nuclease-free PBS and fixed eyecups in 4% paraformaldehyde (PFA) solution in PBS for 24 h at 4 °C[43]. A large relieving cut was made in the nasal margin of the eyecup. For experiments at embryonic stages, WT pregnant females were anesthetized by IP injection of Avertin and sacrificed by cervical dislocation. Embryos were dissected, fixed in PFA 4% overnight. Eyecups and embryos were then washed in PBS and cryoprotected in a 30% sucrose solution in PBS overnight at 4 °C. Tissues were then embedded and frozen in OCT using dry ice. 20 µm sections were obtained on a Leica CM1950 cryostat and mounted directly onto SuperFrost Plus slides (Fisher). The tissue was processed according to the RNAscope Multiplex Fluorescent v2 assay (Advanced Cell Diagnostics) instructions provided by the manufacturer. Probes (Advanced Cell Diagnostics) and fluorophores (TSA Plus Fluorescence kits, Akoya Biosciences) are listed in Supplementary Table 1. The tissue was then incubated in DAPI solution (Sigma) and mounted using ProLong Glass Antifade Mountant (Thermo). All dissections were performed at ZT6-ZT7, and the tissue of all experimental groups was processed at the same time to account for variability between replicates.

Sections were imaged on a Leica SP5 confocal microscope. For quantification, high magnification images (367.38 µm × 367.38 µm with a pixel size of 0.71 µm) with a z-stack size of 1 µm were taken. Because ipRGCs are a sparse population of RGC, regions of interest (ROIs) around *Opn4* mRNA or *mCherry* mRNA (Fig. 2B and Supplementary Figs. 3–6, 8 and 12) were manually drawn. The mean pixel intensity, per ROI, per stack, per probe, was established using ImageJ (RRID:SCR_003070). Analyses were performed on 336 cells of 6 retinas from *Opn4$^{Cre/+}$; Brn3b$^{+/+}$* mice and 213 cells of 6 retinas from *Opn4$^{Cre/+}$; Brn3b$^{cKOAP/cKOAP}$* mice (Fig. 2B), 371 cells of 2 retinas from Vehicle-injected *Opn4$^{CerERT2/+}$ Brn3b$^{cKOAP/cKOAP}$* mice and 288 cells of 2 retinas from TMX-injected *Opn4$^{CerERT2/+}$ Brn3b$^{cKOAP/cKOAP}$* mice (Fig. 2C and Supplementary Fig. 8), 24 cells of 2 retinas from *Opn4$^{Cre/+}$; Brn3b$^{+/+}$* mice (Supplementary Fig. 12). Qualitative analysis was done in 18 retinas from 9 WT embryos (Supplementary Fig. 2).

To perform RNA FISH in whole mount retinas, mice were sacrificed and retinas were dissected and fixed as described above. Retinas were washed in PBS, dehydrated in a graded methanol (MeOH)/PBS series (50%, 75% MeOH-PBS, 100% MeOH) for 5 min each. Retinas were stored at 100% MeOH at -20 C overnight. The tissue was rehydrated in a reverse MeOH/PBS series. Retinas were then incubated with RNAscope Protease Plus Reagent (Advanced Cell Diagnostics) for 30 min at 40 °C and then incubated with probes (Supplementary Table 1) overnight. Retinas were washed and post-fixated for 10 min at room temperature in PFA 4%. The tissue was processed according to the protocol provided by the manufacturer. Retina were mounted and sealed as described above.

For quantification, high magnification images (183.69 µm × 183.69 µm with a pixel size of 0.36 µm) with a z-stack size of 1 µm were taken in a Leica SP5 confocal microscope. ROIs around *Opn4* mRNA and control background were drawn, and the mean of pixel intensity, per ROI, per stack, per probe was established using ImageJ. The mean background was subtracted, and label intensity data for each ROI was normalized against the maximum intensity value observed within that retina. Analyses were performed 182 cells of 2 retinas from *Opn4$^{Cre/+}$; Brn3b$^{+/+}$* mice and 241 cells from 2 retinas from *Opn4$^{Cre/+}$; Brn3b$^{cKOAP}$* mice (Fig. 1I–K and Supplementary Figs. 3, 5). *Zcchc12*-positive (Zcchc12$^+$) cells were considered when mean intensity of ROIs were positive, while *Zcchc12*-negative (Zcchc12-) cells were computed when ROIs presented mean intensity was ≤ 0.

## Immunohistochemical procedures

Retinas were dissected as described above and then fixed in PFA 4% in PBS for 30–60 min at RT. Retinas were washed and then blocked at 4 °C overnight in 6% normal donkey serum in 0.3% Triton PBS prior to incubating in primary antibody solution for 2-3 nights at 4 °C (Supplementary Table 1). Then, retinas were washed, then incubated in secondary antibody solution for 2 h at RT (Supplementary Table 1) and mounted using Fluoromount (Sigma). Primary and secondary antibody solutions were made in 3% normal goat serum in 0.3% Triton PBS.

For quantification, high magnification images (183.69 µm × 183.69 µm with a pixel size of 0.36 µm) with a z-stack size of 1 µm were taken and processed using ImageJ. Total number of cells in the retinal ganglion cell layer (GCL) and the density of cells of the outer nuclear layer (ONL) were quantified in 6 retinas from *Opn4$^{Cre/+}$; Brn3b$^{+/+}$; Rpl22$^{HA}$* mice and 5 retinas from *Opn4$^{Cre/+}$; Brn3b$^{cKOAP/cKOAP}$ Rpl22$^{HA}$* mice (Supplementary Fig. 23). ROIs around melanopsin-immunolabel were manually drawn and the mean pixel intensity, per ROI, per stack, per probe was established. The mean background was subtracted, and label intensity data for each ROI was normalized against the maximum intensity value observed within that retina. Analyses of melanopsin levels were performed in 600 cells of 5 retinas from *Opn4$^{Cre/+}$; Brn3b$^{+/+}$* mice (Fig. 2D–F and Supplementary Fig. 7), 600 cells from 4 retinas from *Opn4$^{Cre/+}$; Brn3b$^{cKOAP/cKOAP}$* mice (Fig. 2D–F and Supplementary Fig. 5), 800 cells of 4 retinas from Vehicle-injected *Opn4$^{CreERT2/+}$; Brn3b$^{cKOAP/cKOAP}$* mice (Fig. 2G, H), 761 cells from 4 retinas from TMX-injected *Opn4$^{Cre/+}$; Brn3b$^{cKOAP/cKOAP}$* mice (Fig. 2G, H). For morphological studies, dendritic arbor tracings were performed using the Simple Neurite Tracer toolbox (https://github.com/morphonets/SNT) for ImageJ in the mCherry channel (Fig. 3B, C; Supplementary Figs. 9–11). ChAT immunolabeling allowed determination of the dendritic stratification level in ON or OFF sublayers of the inner plexiform layer (IPL). Using the intravitreal sparse labeling approach, M4 ipRGCs were identified as mCherry-positive, ON-stratifying, SMI-32-positive (Fig. 3B); while M2 ipRGCs were identified as mCherry-positive, ON-stratifying, SMI-32-negative positive (Fig. 3a and Supplementary Fig. 10). All retrogradely labeled cells from SCN were identified as M1 ipRGCs (mCherry-positive, OFF-stratifying, SMI-32-negative). All morphological parameters were calculated using the Simple Neurite Tracer toolbox.

## Ex vivo retina preparation for electrophysiology

Mice were eye-injected with AAV2-hSyn-DIO-mCherry 1-2 weeks prior to electrophysiology recordings to permit targeted recordings of ipRGCs (Supplementary Fig. 14). 60–150-day old mice were dark-adapted for at least one hour and sacrificed by $CO_2$ asphyxiation. Eyes were enucleated, and the retinas were dissected under dim red light in carbonated (95% $O_2$-5% $CO_2$) Ames' medium (Sigma-Aldrich). Retinas were sliced in half along the nasal-temporal axis and incubated in carbonated Ames' medium in the dark at 26 °C for at least 1 h. Retinas were mounted on a glass-bottom recording chamber and anchored using a platinum ring with nylon mesh (Warner Instruments). The chamber was placed on an electrophysiology rig, and the tissue was perfused with Ames' medium with synaptic blockers. All recordings were made at 25-26 °C.

## Solutions for electrophysiology

All recordings were made in Ames' medium with 23 mM sodium bicarbonate. Synaptic transmission was blocked with 100 µm DNQX (Tocris), 20 µm L-AP4 (Tocris), 100 µm picrotoxin (Sigma-Aldrich), and 20 µm strychnine (Sigma-Aldrich) in Ames' medium. All recordings were made with internal solution containing 125 mM K-gluconate, 2 mM $CaCl_2$, 2 mM $MgCl_2$, 10 mM EGTA, 10 mM HEPES, 10 mM Na$_2$-ATP, 0.5 mM Na-GTP, and 0.3% Neurobiotin (Vector Laboratories).

## ipRGC electrophysiological recordings and data analysis

Recordings were made using borosilicate pipettes (Sutter Instruments) with resistances between 2–6 MΩ for M4 ipRGCs and 4–8 MΩ for M1 and M2 ipRGCs. Data was collected using a Multiclamp 700B amplifier

(Molecular Devices) with pClamp 10 (RRID:SCR_011323) acquisition software. Reported voltages are corrected for 14 mV liquid junction potential between the Ames' medium and K-gluconate internal solution. To identify ipRGCs, mCherry or YFP-expressing cells were visualized with epifluorescence and M4, M2, and M1 ipRGCs were targeted based on soma size. A blue LED light ($\sim$ 480 nm) was used to deliver light stimuli to the retina through a $\times 60$ water-immersion objective. The photon flux was attenuated using neutral density filters (Thor Labs).

To measure input resistance, cells were voltage clamped at $-70$ mV and hyperpolarized with a $-10$ mV step. Input resistance was calculated with Ohm's law using the steady-state current during the voltage step. To measure the excitability, cells were held at approximately $-70$ mV in current clamp and then 0.5 s current injections ramping from 50–400 pA in M4 ipRGCs, 25–200 pA for M2ipRGCs and 10–80 pA for M1 ipRGCs were applied. To account for current density, we normalized the current injections to cell size by dividing by the whole cell capacitance.

To calculate the membrane area for each ipRGC type, we divided the whole-cell capacitance by the specific membrane capacitance of $0.9$ pF $\cdot$ cm$^{-2}$, as defined in ref. [44] (Supplementary Fig. 15). To determine the specific input resistance, we then divided the input resistance by the membrane area (Supplementary Fig. 15).

To measure the intrinsic light response, cells were voltage clamped at $-70$ mV and the photocurrent was recorded following a 50 ms full-field flash of bright light with an intensity of $6.08 \times 10^{15}$ photons $\cdot$ cm$^{-2}$ $\cdot$ s$^{-1}$. Only one light response was recorded per retina piece to ensure that light adaptation was consistent across recordings. The Maximum photocurrent represents the greatest change from baseline after light onset, and the Sustained photocurrent was calculated as the average change from baseline during the 40–50 s after light onset. Photocurrent values were reported as the absolute value of the current for each component. Electrophysiological data was analyzed using the Python Programming Language v3.12.7 (RRID: SCR_008394) and the PyABF package (https://pypi.org/project/pyabf/) and custom scripts. The code is available on GitHub (https://github.com/schmidtlab-northwestern).

To identify the recorded neurons, tissue was fixed overnight in 4% paraformaldehyde solution in PBS at 4 °C. Retinas were then washed in PBS for 45 min and blocked overnight in 6% normal donkey serum in 0.3% Triton-X. Retinas were then incubated in primary and secondary antibodies as described above. Sections were then washed and mounted on glass slides. Cells were defined as M4 ipRGCs based on dendritic stratification in ON sublamina of the IPL and SMI32-positive; M2 ipRGCs based on dendritic stratification in ON sublamina of the IPL and SMI32-negative; M1 ipRGCs based on dendritic stratification in OFF sublamina of the IPL; and the presence of an intrinsic light response.

## Optokinetic tracking response

Optokinetic tracking response experiments were performed between Zeitgeber times (ZT) 6 and 8. Spatial frequency thresholds were assessed using the virtual OptoMotry system (Cerebral Mechanics, Lethbridge, Alberta)[12,43,45]. Unrestrained animals were placed individually on an elevated platform at the epicenter of the arena. Animals were presented with stimuli of increasing spatial frequencies at 100% contrast to assess the spatial frequency threshold of the animal in both clockwise and counterclockwise directions[46]. Contrast thresholds measured at 4 spatial frequencies (0.031, 0.064, 0.092, 0.272 c/d), and a contrast sensitivity function was generated by calculating a Michelson contrast using the screen luminance (max-min)/(max+min). Experimenters were blind to an animal's experimental condition and previously recorded thresholds, and adjunct observers were used to validate threshold assessments. All measurements were conducted using a cylinder rotation of 12°/sec, and these thresholds were measured for each eye and averaged for each mouse.

## Voluntary wheel running behavior

Wheel running activity was recorded by individually housing P90 male mice in cages with a running wheel. Activity was recorded using ClockLab Data Collection software (Actimetrics). Mice were first exposed to a 12:12 light-dark (LD) cycle with 100 lux light during the light phase for 3 weeks. Then, animals were exposed to a 6-hour phase advance. Mice that stopped running during the experiment were excluded. All data analysis was performed using ClockLab 6 analysis software (Actimetrics). Circadian amplitude, total activity, percent activity in light and onset error were measured in the 8 days preceding phase advance.

## Pupillometry

For the pupillary light reflex (PLR), experiments were performed between ZT 6-8. Consensual PLR was measured by delivering a 470 nm light stimulus to one eye while simultaneously recording the other eye using a Sony Handycam camcorder. An LED light source was band passed using a 470 nm bandpass filter (Thorlabs), and light intensity was attenuated using neutral density filters (Thorlabs). Mice were dark-adapted for at least 30 minutes prior to any light exposure and were then manually restrained by hand. A 2–5 s baseline recording was taken prior to delivering the 5 s light stimulus. The order in which animals were tested after dark adaptation was randomized. Maximum pupil area was quantified *post hoc* using the oval tool in ImageJ. Steady-state pupil area was then calculated by averaging the pupil area during the last 3 s of the stimulus. To measure the time course of PLR, videos were analyzed using DeepLabCut (RRID:SCR_017302), and data were fitted using a single exponential decay function to measure the time constant tau.

## Statistical comparisons

All graphs and statistical analyses were performed using Graph Pad Prism 10.1.2 (RRID: SCR_002798), Matlab (RRID:SCR_001622), or R (RRID:SCR_001905). Normal distribution of data was tested using the Shapiro-Wilk test. When comparing two groups, Student's t- or Mann-Whitney U tests were used. For multiple statistical comparisons, we performed two-way ANOVA with repeated measures followed by Tukey's *post hoc* test. Significance was concluded when $P < 0.05$. Sex was not considered as a biological variable in the study design or analysis. Data were not disaggregated by sex, as the experiments were not powered to detect sex differences.

## Reporting summary

Further information on research design is available in the Nature Portfolio Reporting Summary linked to this article.

## Data availability

TRAP-seq data are available in the Gene Expression Omnibus (GEO) database under the reference number GSE274888. Mouse RGC snRNAseq and BRN3B CUT&Tag data were downloaded through the GEO repository under the accession codes: GSE137400 and GSE220587. Source data are provided in this paper.

## Code availability

The code for the electrophysiology analysis is available at https://github.com/schmidtlab-northwestern.

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

## Acknowledgements

National Institutes of Health grant R01 EY034662-01A1 (T.M.S. and Y.Y.), National Institutes of Health grant T32 EY025202, National Institutes of Health grant DP2 EY027983 (T.M.S.), National Institutes of Health grant R01 NS123285 (Y.Y.) and National Institutes of Health grant U01 DA053691 (T.Y.). We thank Dr. Samer Hattar for the gift of *Opn4*$^{Cre/+}$ and *Opn4*$^{CreERT2/+}$ mice.

## Author contributions

Conceptualization: M.L.A., Y.Y., and T.M.S. Methodology: M.L.A., J.D.B., O.A.P.P., S.K.L., and T.Y. Investigation: M.L.A., J.D.B., O.A.P.P., T.Y., Y.Y., and T.M.S. Visualization: M.L.A., J.D.B., T.Y., Y.Y., and T.M.S. Funding acquisition: Y.Y. and T.M.S. Project administration: T.M.S. Supervision: Y.Y. and T.M.S. Writing – original draft: M.L.A. and T.M.S. Writing – review & editing: M.L.A., J.D.B., O.A.P.P., T.Y., Y.Y., and T.M.S.

## Competing interests

J.D.B. is a co-founder of Aura Life Science, in which he has a financial interest. All other authors declare that they have no competing interests.
