## [Transparent Peer review file · Nature Communications]

Genetic tuning of retinal ganglion cell subtype identity to drive visual behavior

Corresponding Author: Dr Tiffany Schmidt

Version 0:

Reviewer comments:

Reviewer #1

(Remarks to the Author)

Aranda et al. is an interesting paper that describes how the loss of Brn3b/Pou4f2 impacts various physiological and morphological characteristics in different ipRGC subtypes. This work provides new insights into how the dosage of a single transcription factor may contribute to maintaining the identity of different ipRGC subtypes. The study is well-planned, executed, and presented. However, the following concerns should be addressed to avoid misinterpretation.

The major concern is whether Brn3b was completely deleted in all Opn4+ cells and whether all mutant cells survived without Brn3b, as the authors claimed. If not, their subsequent recording and analysis may be problematic, if not flawed. The evidence presented in the manuscripts was: 1) RNA-seq on ribosome-bound RNAs using anti-HA-tag pull-down, suggesting the presence of Opn4-Cre activated Rpl22-HA ribosomes; 2) A very selective image (Fig. 2B and Extended Fig. 3) showing some melanopsin+ cells do not express Brn3b by ISH; and 3) Wholemout IF data (Extended Fig. 5) showing the number of melanopsin+ cells was unaffected in the mutant. This evidence is insufficient to conclude that Brn3b was deleted in all Opn4+ cells and that all mutant cells without Brn3b survive. The following experiments should be included: 1. Conduct co-immunolabeling (as shown in Extended Fig. 5) on control and mutant retinal flatmounts using melanopsin, Brn3b, and HA antibodies together to verify whether Brn3b was indeed deleted in all Opn4+ cells. If the result holds up and all mutant cells survive, the number of HA+ cells should be similar between the control and mutant, and Brn3b expression should be undetectable in the mutant cells. 2. Conduct alkaline phosphatase staining between the control (Opn4-cre; Brn3b-CKOAP/+) and mutant (Opn4-cre; Brn3b-CKOAP/CKOAP). The number of AP+ soma should be very similar between the two groups.

A similar concern arises with the iBrn3b experiment in Extended Fig. 6D: while the average Brn3b mRNA levels were reduced in iBrn3bKO, many mutant cells still contained high levels of Brn3b. This is odd because iBrn3b is a conditional KO allele in which the Brn3b gene should be deleted, and no signal should be detected. A proper experiment would involve conducting ISH (similar to that in Extended Fig. 3C) using anti-Brn3b, Opn4, and PLAP (the reporter gene in the Brn3b-CKOAP allele) probes together. This way, the authors can determine whether both Brn3b-CKOAP alleles were deleted, and the AP+ cells in iBrn3b should not express Brn3b at all (one possible scenario: one allele was deleted, but the second allele was not).

Minor points:

- The affected genes described in Fig. 1A should be cross-referenced with known Brn3b targets (Ge et al., Nucleic Acids Research, doi: 10.1093/nar/gkad026), and the complete list should be presented in an Excel file.
- In methods: please list the Geo dataset number in Methods for scRNA seq reanalysis (Tran et al., 2019); describe the criteria for how ipRGCs were subsetted and what markers were used to annotate ipRGC subtypes.
- In Fig. 2C, there appears to be a significant difference in Opn4 mRNA levels (measured by pixel intensity) between the two experimental settings (developmental and inducible). Why is that?
- In line 143, it states that '~10% of Brn3bcKO M2/M4 cells stratified to the OFF sublamina.' This observation is very interesting, and 10% is significant. A few representative images should be presented somewhere, ideally with a reference marker such as Chat.

Reviewer #2

(Remarks to the Author)

This manuscript examines the role of the transcription factor Brn3b in regulating aspects of ipRGC function and behavior. The experiments illustrate the differential expression of Brn3b across ipRGC subtypes by using available scRNA seq datasets and FISH. The authors hypothesize that differential Brn3b expression underlies ipRGC subtype identity and shapes each subtypes differential morphology and physiology. Experiments were motivated by observations of ipRGC morphology/complexity/subtypes in relation to Brn3b expression. Genetic markers are used to distinguish subtypes of RGCs so it is important to fully understand their contribution to subtype specificity. The authors use an impressive array of tools including IHC, FISH, transgenic mice, patch-clamp electrophysiology, RNA sequencing, and behavior to approach this question. The paper contains valuable morphological and physiological measurements of commonly used tools in the field. However, there are several major issues with the experiments, data interpretation and there is a lack of mechanistic ties between the functional changes in ipRGCs in the retina and behavioral results which need to be clarified by more experiments.

The noteworthy results of this work are:

1. The finding that all M1 ipRGCs express Brn3b as it was previously thought there is a Brn3b-negative population of M1 ipRGCs in the mouse retina.
2. Brn3b deletion results in changes in the expression of Opn4 mRNA and melanopsin protein.
3. Brn3b deletion alters the morphological properties of ipRGCs, which distinguish many ipRGC subtypes from each other in the mouse retina.
4. Brn3b deletion changes the excitability of M4 ipRGCs.

Major Comments

1. The authors use TRAPseq to sequence Opn4Cre expressing neurons in the retina and illustrate that ipRGC-specific mRNA are up or downregulated when Brn3b is deleted. High Brn3b expressing neurons (M4-6) show downregulation of specific genes, indicating that Brn3b is important for the transcriptional regulation of these genes. Low Brn3b-expressing ipRGCs (M1-3) show upregulation of some ipRGC-specific genes, illustrating that Brn3b may be involved in the transcriptional suppression of these genes. A mechanistic description of how Brn3b might underlie the subtype identity of ipRGCs is unclear, and it is also unclear how the Brn3b-dependent regulation of these transcripts impacts cellular diversity or function. At present, there is only a descriptive tie between Brn3b deletion and changes in other genes. Given the results in the later parts of the manuscript indicating Opn4 mRNA and melanopsin protein are upregulated in Brn3bcKO mice, it is confusing that Opn4 gene expression is not included in the TRAPseq analysis.
2. Given the OPN4Cre-dependent excision of Brn3b occurs “well after post-mitotic ipRGC specification” and there appears to be limited differences in the most pronounced subtype-specific characteristics such as: melanopsin protein concentrations, melanopsin response (limited analysis to M4 ipRGCs), dendritic stratification, central projection targets (only minimally examined with retrograde injections into the SCN), how can the authors rule out the hypothesis that subtype specification (M1-M6) occurs during post-mitotic specification and before their Brn3b deletion?
3. Next, the manuscript examines the Brn3b regulation of melanopsin expression in Brn3bcKO mice indicating that Brn3bcKO increases Opn4 mRNA expression and melanopsin protein levels. These data appear to support the conclusion that Brn3bcKO alters Opn4 mRNA and melanopsin protein levels, but this is confusing because if the hypothesis is that Brn3b suppresses Opn4 and melanopsin then one would expect that the Opn4 mRNA and melanopsin protein would be particularly high in the M4-M6 ipRGCs following Brn3b deletion, which is not clear from the illustrated data. The example images shown in Fig. 2D seem to illustrate some weak staining of at least one low Brn3b ipRGC in the control (weak and large soma in the middle) whereas this is not present in the Brn3bcKO. Which areas of retina are these examples taken from, dorsal or ventral retina? Perhaps these images could be reexamined to determine if the larger number of ipRGCs expressing high levels of melanopsin are putative M1 or M2 ipRGCs and some quantification of their soma size might shed light on this.
4. Next the manuscript examines the impact of Brn3b expression on the morphology of ipRGCs using Brn3bcKO mice and intravitreal injections of Cre-dependent AAV expressing mCherry to label M2 and M4 ipRGCs and retrograde injections into the SCN to label M1 ipRGCs. Brn3bcKO clearly has an effect on reducing the dendritic arbor size of ipRGCs, but it is a bit of a stretch to say that reduced arbor size and reduced arbor complexity pushes the M2-M4 ipRGCs towards a M1 morphological phenotype. Some other clarification is needed in this section (lines 136-146) as the authors state that “M1 cells are not labeled at this low titer” but also mention ~10% of ipRGCs stratified in the OFF sublamina. Are these M1 ipRGCs, or are they putative M2 and M4 ipRGCs that have been mis-stratified? Dendritic stratification is one of the most defining characteristics of the M1 ipRGCs. Are these OFF stratifying ipRGCs SMI-32 positive or negative? The results of these data suggest that Brn3b controls aspects of soma and dendritic growth, but it is unclear from the presented data how Brn3b defines the morphological identity of these specific subtypes.
5. Next, the manuscript describes the impact of Brn3bcKO on the electrophysiological properties of M4 ipRGCs. M4 ipRGCs in Brn3bcKO mice showed increased input resistance and increased excitability when compared to controls and were more prone to go into depolarization block to high amplitude current injection. The intrinsic photocurrent of M4 ipRGCs however, was unaffected by Brn3bcKO. It is difficult to reason how Brn3b deletion is causing the changes in M4 input resistance and

excitability outside of the smaller soma size and dendritic size. The authors accounted for current density but do not detail how the current density was accounted for or calculated. If there is a Brn3b-mediated change in the input resistance and excitability and it does not relate to the changes in morphology, there is no mechanism proposed for how this occurs. Is this due to changes in the density of ion channels? Is there any more evidence to better support this claim?

6. Previously, the authors claimed that Brn3b controls the level of Opn4 mRNA and melanopsin expression, but the previous figures don't address the melanopsin expression in M4 ipRGCs. The rationale for examining the electrophysiological changes in M4 ipRGCs in the Brn3bcKO mice is that M4 ipRGCs have the highest levels of Brn3b expression. Given that the melanopsin expression is not examined in the M4 ipRGCs and that there is no change in the intrinsic photocurrent in M4 ipRGCs with Brn3bcKO doesn't this suggest that Brn3b has little impact on melanopsin expression? The physiological data together with the lack of clear RNA and immunofluorescence data from this cell type does not support this main hypothesis. Would it be possible to examine the melanopsin protein levels in the ipRGCs examined in Figure 3?

7. If Brn3b deletion increases the melanopsin expression in ipRGCs (M1 and M2?) there should be physiological data to support this by measuring the intrinsic photocurrents in the ipRGCs that have higher melanopsin expression. It is unclear why the mRNA expression and melanopsin protein expression is examined in detail (Figure 2) but not backed up by electrophysiology to illustrate the physiological impact of this expression change in ipRGCs other than M4s.

8. Next, the manuscript details the impact of Brn3bcKO on various ipRGC-mediated behavioral outcomes. First, they examine the contrast sensitivity of the optokinetic tracking response, which they claim is mediated by M4 ipRGCs. If Brn3b reduces the contrast sensitivity of the OKT through differences in M4 function, then the authors should strengthen this finding by measuring the contrast sensitivity of the M4 ipRGCs in the Brn3bcKO mouse retinas. The increased excitability and higher input resistance of these cells without any appreciable differences in cellular function until high levels of current injection would predict the opposite effect where lower levels of contrast are more likely to drive spike output from M4 ipRGCs. You would therefore predict that Brn3bcKO mice would have higher contrast sensitivity. Could this effect be mediated through impacts on photoreceptors, or through Brn3b-deletion in central neurons, many of which express OPN4 in the OPN4Cre(saha) mouse line (PMID: 39127043)?

9. Similarly, the impairment of the bright PLR might arise through differential central mechanisms introduced by Brn3b-deletion and this result is also a bit confusing. This is because, Brn3b deletion is supposed to increase the melanopsin expression in M1 ipRGCs and one might expect that the increased melanopsin expression would enhance the bright light PLR, which is primarily mediated by the intrinsic photosensitivity of M1 ipRGCs. Is there a mechanism proposed for how Brn3bcKO might impact the bright PLR through decreased M1 activity in the retina? Recording from M1 ipRGCs in Brn3bcKO retinas might provide some insight into how this might occur. Similarly, issues with the direct mechanistic link between ipRGC dysfunction and circadian entrainment are not strong without direct recordings to confirm that Brn3bcKO is impacting the physiology or axon targeting of M1 ipRGCs. These complex behavioral effects might arise from the deletion of Brn3b from central targets using this Cre line (PMID: 38659888).

Minor comments:

1. The Opn4Cre(saha) mouse has ectopic labeling in amacrine cells of the retina potentially confounding TrapSeq experiments (PMID: 39127043). Are there any controls to determine the ssRNAseq data excludes retinal amacrine cells?

2. Many supplemental figures can be combined with main figs of which there are only 4. Figure 3 could be split into two figures, one for morphology and one for physiology. It would be nice to show the embryonic data in a main text figure.

3. Was the FISH processing of control and experimental retinas performed at the same time to account for variability between replicates?

4. Were the eyes collected at the same time of day given that Opn4 expression varies across time of day?

5. Why is there Ca²⁺ in the internal solution? 2mM seems quite high.

6. Cells were held at -70mV in current clamp. What are the values of the holding current used to keep cells at -70mV and what was their resting membrane potential? This is important because if the cKO m4 ipRGCs have different resting membrane potential values this might impact their excitability.

7. Were experimental mice homozygous for Brn3bcKOAP? Or were the heterozygous/have a normal Brn3b allele?

8. It would be useful to report the top 50 differentially expressed genes and associated p-values from Opn4Cre vs total retina and control vs Brn3b KO experiments.

9. Contrast levels are not detailed for the contrast sensitivity experiments.

10. Gene names should be italicized and protein names in all caps to clearly distinguish gene and protein names/labels.

11. Please use melanopsin instead of "Opn4 protein" or "anti-Opn4". OPN4 should be capitalized if referring to protein but using Opn4 for the gene and melanopsin for the protein is less confusing.

12. Please report the actual genotype instead of "control" in the text. I.e. Opn4Cre^{+/+};Brn3bcKOAP^{-/-} or include a statement

that control animals are *Opn4Cre^{+/+};Brn3bcKOAP^{-/-}* mice.

13. Line 64/65 – How does Fig 1 B/C show graded expression of transcripts?
14. Line 86 – should *Brn3b* KO lead to increased *Chrna6* expression in only M4-M6?
15. Line 103 – genotype of control and experimental mice should be clearly defined in text and figure legend
16. Line 107 – which types of ipRGCs are these?
17. Line 162 – ventral M1 ipRGCs are thought to have the highest melanopsin expression (PMID: 23954426 PMID: 28965762) so, one would expect them to have less, not more, *Brn3b* than dorsal M1s.
18. Line 165 – should say extended fig. 8.
19. Line 166 – Please clarify this is mRNA and not protein (“modest amounts of *Brn3b*”).
20. Line 467 – Goat serum listed but donkey serum listed in line 539
21. Line 705 – What are the established genetic markers used to establish annotate ipRGC subtypes?
22. Extended data Fig. 7 – typos in panel C/D “lengh”
23. Line 761 – should “Ventral....higher levels...than...ventral...(change one to dorsal).

Reviewer #3

(Remarks to the Author)

In this manuscript Aranda et al. use a combination of techniques that all support the conclusion that the levels of expression of *Brn3b* mRNA regulate the expression of melanopsin in ipRGCs as well as the differentiation into different ipRGC types. There are 6 ipRGC types that differ in their projection patterns, morphological properties, and melanopsin expression (leading to differences in light response properties). It was originally reported that M1 ipRGCs which have the highest melanopsin expression are *Brn3* negative, while the other ipRGC types have *Brn3b* expression. More recent RNA profiling and verified here, shows that *Brn3b* is expressed in M1 ipRGCs but the levels are decreased compared to the other ipRGC types. The authors show that deletion of *Brn3b* in non-M1 ipRGCs leads to altered transcriptional patterns of genes expressed in ipRGC subtypes including an increase in *OPN4* expression and this is associated with changes in dendritic morphology, and intrinsic properties toward a M1 fate. Showing that *Brn3b* is an important regulator of ipRGC subtype differentiation. Overall, the results are well controlled and compelling and offer mechanistic insight into how changes in gene transcription can regulate subtle changes between cell types. This work will be of high interest to those studying the development of the retina, and those studying how levels of TF regulate subtype specificity.

I have a few questions/comments that should be clarified/addressed that will help me understand the paper better.

1. According to the Broad single cell RGC data base https://singlecell.broadinstitute.org/single_cell/study/SCP509/mouse-retinal-ganglion-cell-adult-atlas-and-optic-nerve-crush-time-series?genes=Opn4%2CPou4f2%2CChrna6%2CZcchc12&cluster=Atlas RGCs&spatialGroups=-&annotation=Cluster--group--cluster&subsample=all&tab=dotplot#study-visualize *Brn3b* is not restricted to ipRGCs, it is expressed in a large number of non-ipRGC types. Does *Brn3b* regulate *Opn4* expression in these types? The authors only remove it from ipRGCs and only assay changes in ipRGCs but in my mind it could/should be repressing *OPN4* in these other types that are not photosensitive.
2. *Chrna6* and *Zcchc12* expression is not restricted to ipRGC. What happens to *Chrna6* and *Zcchc12* expression when *Brn3b* is removed from non-ipRGCs.
3. Do *Brn3b* levels decrease by ½ in heterozygous mice? If so I wonder if this changes the numbers of ipRGC types. In other words does removing ½ of the expression push cells towards the M1 fate?
4. Currently the evidence for a graded model is that by removing expression leads to an increase in M1 characteristics, but the model also predicts that an increase of *Brn3b* expression in M1 would direct them to a M4-6 fate, or even a non-ipRGC fate. It would strengthen the argument if this was shown.
5. I worry a bit about the use of “gradient” used in abstract and discussion. I am not clear what it means in this context, high, medium and low levels? Can one really quantify protein expression levels based on RNA scope results to differentiate between discrete steps vs. a gradient of expression?
6. The methods also do not determine whether *Brn3b* directly or indirectly regulates *Opn4* expression. Chip seq experiments using *Brn3b* are needed to distinguish between these possibilities. This should be stated in the discussion if this experiment is deemed too much work for this report.
7. There is no mention of how this result reconciles with studies that show that *eomes/Tbr2* is essential for the expression of melanopsin when removed after development. Was *eomes* found to change expression in *Brn3b* mutants or are there two

district pathways to get to OPN4?

8. Line 42 could be more clear about the expression “Notably, Brn3b expression is present in newly postmitotic ipRGCs and persists into adulthood, suggesting it may play yet unidentified roles in ipRGC development and function” as stated above Brn3b is also expressed in a number of non-ipRGCs.

9. Line 243 discussion should be modified. The RGC RNA seq atlas of RGCs shows this to be true.

“Surprisingly, we find that even SCN-M1 ipRGCs show low levels of Brn3b expression that are critical for shaping cellular properties and their role in circadian photoentrainment, highlighting 245 its critical role in defining the full spectrum of ipRGC subtypes.”

10. Extended data 2: what are the arrows pointing to? Please add in figure legend.
Where are the D-V, N-T axes of the retinas?

11. Can the authors relate the levels of a transcription factor to other instances where levels have been shown to control differentiation? Is this unique to ipRGCs is this a common phenomenon?

Reviewer #4

(Remarks to the Author)

Version 1:

Reviewer comments:

Reviewer #1

(Remarks to the Author)

The authors have thoroughly responded to my comments, and I have no further questions.

Reviewer #2

(Remarks to the Author)

I thank the authors for making detailed changes to the manuscript including additional experiments and analysis. The manuscript is greatly improved, and I have no further requests.

Reviewer #3

(Remarks to the Author)

The authors have done their best to answer my and the other reviewer's questions and it is ready for publication.

Reviewer #4

(Remarks to the Author)

We are resubmitting a revised version of our work entitled “Genetic tuning of retinal ganglion cell subtype identity to drive visual behavior” for consideration. We thank the reviewers for their thoughtful and in-depth reviews of the initial submission of this manuscript. In this study, we identify a transcriptional mechanism that generates the distinct genetic and morphophysiological properties of multiple neuronal subtypes in the retina and is essential for their unique roles in diverse visual behaviors. Broadly speaking, we have added multiple new approaches to test the regulatory role of Brn3b on physiological properties as well as melanopsin and other gene expression levels in specific ipRGC subtypes to address the reviewer concerns. Here we list the additions to the manuscript and have responded to the specific comments of each reviewer in the body of the text.

- Analysis of BRN3B Cut &Tag, scATAC-seq¹ and Brn3bcKO downregulated genes in TRAPseq datasets.
- Analysis of RNAscope - Density of *Opn4* mRNA positive cells in Control and Brn3bcKO retinas.
- Analysis of RNAscope - *Opn4* and *Chrna6* mRNA levels in M4, M2 and M1 ipRGC binning by soma size.
- RNAscope with immunolabeling in flat mount retinas: *Opn4*, *Brn3b* mRNA probes and anti-Calbindin antibody in Control and Brn3bcKO mice.
- RNAscope in flat mount retinas: *Opn4* and *Brn3b* mRNA probes in Vehicle- and TMX-injected iBrn3bcKO mice.
- RNAscope in retina sections: *Opn4*, *Brn3b* and *Eomes* mRNA probes.
- Representative images from our anti-melanopsin immunolabeling from all quadrants of Brn3bcKO and control littermates
- Representative images of M4 and putative M2 ipRGCs with OFF-stratifying dendrites
- Intrinsic electrophysiological properties in M2 and M1 ipRGCs from Control and Brn3bcKO retinas.
- Photocurrent responses in M2 and M1 ipRGCs from Control and Brn3bcKO retinas.
- Maximum contrast sensitivity levels in OKR experiments in Control and Brn3bcKO mice.
- List of differentially expressed genes and p-values from TRAPseq dataset.
- RNAscope in flat mount retinas: *Opn4* and *Brn3b* mRNA probes in Control and Heterozygous Brn3bcKO (*Opn4*^{Cre/+} *Brn3b*^{CKOAP/+})

We are happy to report that these new results provide further support our original conclusions and greatly strengthen the paper. These findings will truly shift our framework of understanding the role for a gradient of a single transcription factor in tuning the identity of multiple neuronal subtypes, and we hope that the reviewers will appreciate the multiple additional supporting experiments and controls in this updated version.

Reviewer 1

Aranda et al. is an interesting paper that describes how the loss of Brn3b/Pou4f2 impacts various physiological and morphological characteristics in different ipRGC subtypes. This work provides new insights into how the dosage of a single transcription factor may contribute to maintaining the identity of different ipRGC subtypes. The study is well-planned, executed, and presented. However, the following concerns should be addressed to avoid misinterpretation.

The major concern is whether Brn3b was completely deleted in all *Opn4*⁺ cells and whether all mutant cells survived without Brn3b, as the authors claimed. If not, their subsequent recording and analysis may be problematic, if not flawed. The evidence presented in the manuscripts was: 1) RNA-seq on ribosome-bound RNAs using anti-HA-tag pull-down, suggesting the presence of *Opn4*-Cre activated Rpl22-HA ribosomes; 2) A very selective image (Fig. 2B and Extended Fig. 3) showing some melanopsin⁺ cells do not express Brn3b by ISH; and 3) Wholemout IF data (Extended Fig. 5) showing the number of melanopsin⁺ cells was unaffected in the mutant. This evidence is insufficient to conclude that Brn3b was deleted in all *Opn4*⁺ cells and that all mutant cells without Brn3b survive.

1. Conduct co-immunolabeling (as shown in Extended Fig. 5) on control and mutant retinal flat mounts using melanopsin, Brn3b, and HA antibodies together to verify whether Brn3b was indeed deleted in all *Opn4*⁺ cells. If the result holds up and all mutant cells survive, the number of HA⁺ cells should be similar between the control and mutant, and Brn3b expression should be undetectable in the mutant cells.

In order to determine the **efficiency of Brn3b deletion in *Opn4*⁺ cells in the Brn3bcKO line**, we performed RNA scope labeling of *Opn4* and *Brn3b* mRNA, which allows for highly sensitive detection of transcripts for both *Opn4* and *Brn3b*. We chose to do these experiments in flat mount retinas in order to ensure sampling of ipRGCs

from all retinal quadrants. Our data indicate that just 7.4% of *Opn4*⁺ cells in the *Brn3bcKO* line also express *Brn3b*, while 73.5% of *Opn4*⁺ cells in control littermates express *Brn3b* mRNA, indicating that BRN3B expression is lost in the vast majority of ipRGCs in *Brn3bcKO* retinas (Extended Data Figure 3E). The remaining *Brn3b* expression in less than 10% of ipRGCs in the *Brn3bcKO* line is likely due to the fact that Cre recombination is not 100% efficient in transgenic lines²⁻⁴. Some remaining *Brn3b* expression is to be expected based on known recombination rates in Cre transgenic lines and is very low in this case. Importantly, the remaining *Brn3b* expression in this subset of ipRGCs would, if anything, dampen the magnitude of any observed phenotypes, further supporting a key role for *Brn3b* in tuning ipRGC subtype properties. We thank the reviewer for raising this important point.

We address ipRGC survival in our response to #2 below.

2. Conduct alkaline phosphatase staining between the control (*Opn4-cre*; *Brn3b-CKOAP/+*) and mutant (*Opn4-cre*; *Brn3b-CKOAP/CKOAP*). The number of AP⁺ soma should be very similar between the two groups.

In order to **compare the number of ipRGCs in the *Brn3bcKO* versus control littermate retinas**, we used RNA scope labeling of *Opn4* mRNA to quantify the total density of *Opn4*⁺ cells and found **no difference between control and *Brn3bcKO* littermates**. This supports our melanopsin immunolabeling results (Extended Data Figure 5C) from the original submission, which indicated that ipRGC numbers are not reduced in the *Brn3bcKO* line. We further quantified the number of M4 ipRGCs in *Brn3bcKO* and control littermate retinas by using RNAscope to label for *Opn4* and Calbindin (*Calb1*) mRNA. Calbindin is expressed in M4 ipRGCs but not in other ipRGC subtypes. Thus, *Opn4* and *Calb1* double positive cells represent M4 ipRGCs. We found **no reduction in the overall density of M4 cells in *Brn3bcKO* retinas**. Interestingly, M4 spatial distribution shifted from being temporally enriched to more evenly distributed across the retina⁵⁻⁷. When we compared the densities of **non-M4** ipRGCs, we again **found no significant difference**. Collectively, these findings indicate that all ipRGC subtypes survive and are present in similar numbers in *Brn3bcKO* retinas (Extended Figs. 5, 7).

The similar total number of each ipRGC subtype, suggests that there was not loss of specific subtypes in the absence of *Brn3b*. Notably, we did report a reduced number of HA⁺ cells in *Brn3bcKO* retinas using the Ribotag (*Rpl22^{HA}*) line in our original submission. However, given the added recombination and reporter expression steps necessary to label cells with HA, we are inclined to weigh results from the direct *Opn4* labeling more heavily, and we now address this in the Discussion. We chose not to perform alkaline phosphatase labeling as it relies on enzymatic amplification that could be skewed by copy number and incubation time. Moreover, *Opn4* provides a more direct quantification of ipRGC numbers.

3. A similar concern arises with the *iBrn3b* experiment in Extended Fig. 6D: while the average *Brn3b* mRNA levels were reduced in *iBrn3bcKO*, many mutant cells still contained high levels of *Brn3b*. This is odd because *iBrn3b* is a conditional KO allele in which the *Brn3b* gene should be deleted, and no signal should be detected. A proper experiment would involve conducting ISH (similar to that in Extended Fig. 3C) using anti-*Brn3b*, *Opn4*, and PLAP (the reporter gene in the *Brn3b-CKOAP* allele) probes together. This way, the authors can determine whether both *Brn3b-CKOAP* alleles were deleted, and the AP⁺ cells in *iBrn3b* should not express *Brn3b* at all (one possible scenario: one allele was deleted, but the second allele was not).

In order to determine the **efficiency of *Brn3b* deletion in *Opn4*⁺ cells in the *iBrn3bcKO* line**, we performed RNA scope labeling of *Opn4* and *Brn3b* mRNA, which allows for highly sensitive detection of transcripts for both *Opn4* and *Brn3b*. We performed these experiments in flat mount retinas of vehicle- or TMX-injected *iBrn3bcKO* littermates. While vehicle-injected *iBrn3bcKO* retinas showed 77.8% of *Opn4*⁺ cells expressed *Brn3b*, we found that just 31.3% of *Opn4*⁺ cells express *Brn3b* in TMX-injected *iBrn3bcKO* retinas (Extended Data Figure 8F). The lower recombination efficiency is to be expected with the tamoxifen-inducible system that relies on TMX injection, bioavailability, and efficiency of driving Cre expression from the relatively weak *Opn4* promoter⁸⁻¹⁰. Nonetheless, our data indicate a reduction in *Brn3b*⁺ ipRGCs. Extended Data Figure 8E also shows that the remaining cells expressed lower levels of *Brn3b*.

Thus, our findings show that we detect BRN3B-dependent phenotypic changes *despite* a lower recombination efficiency in the *iBrn3bcKO*s, which could account for the overall milder phenotype. We now note this in the Discussion section.

Minor points:

a. The affected genes described in Fig. 1A should be cross-referenced with known *Brn3b* targets (Ge et al., *Nucleic Acids Research*, doi: 10.1093/nar/gkad026), and the complete list should be presented in an Excel file.

As suggested, we downloaded **BRN3B Cut & Tag data from E14.5 retina¹ and analyzed it together with our *Brn3bcKO* TRAP-seq data from adult ipRGCs**. We find that BRN3B binding is enriched at genomic regions overlapping with genes that are downregulated, but not upregulated, upon *Brn3bcKO* (Extended Data Figure 4). We note that this overlap is likely an underestimate because many genes expressed in adult ipRGCs are absent E14.5 retina¹ (comparing scRNA-seq datasets¹¹), and thus BRN3B binding sites for these adult-specific genes are also missing in E14.5 retina. Overall, these findings suggest that BRN3B directly binds target genes for activation, while it negatively regulates other genes independently of BRN3B binding. We have included the list of BRN3B direct target genes with shared expression between E14.5 retina and adult ipRGCs in the new Extended Data Table 2 in the revised manuscript.

b. In methods: please list the Geo dataset number in Methods for scRNA seq reanalysis (Tran et al., 2019); describe the criteria for how ipRGCs were subsetted and what markers were used to annotate ipRGC subtypes.

We have added GSE137400¹¹ in the Methods section. We obtained ipRGC subtype and cluster annotations from the original study, which were described as M1 (C33/C40), M2 (C31), M4 (C43), and an unclassified subtype (C22). Moreover, C7 and C8 were included given their similarity in gene expression profiles to the M1-M4 ipRGC subtypes, as also noted in recent studies^{12,13}. We denote these as M6 (C7) given its enrichment for *Cdh3*¹⁴ and provisionally as M5 (C8) given its enrichment for a combination of PixON transcripts including *Gnas* derived from Patch-seq analyses^{15,16}.

c. In Fig. 2C, there appears to be a significant difference in *Opn4* mRNA levels (measured by pixel intensity) between the two experimental settings (developmental and inducible). Why is that?

We caution against side-by-side comparisons across these two lines. Differences are likely due to variation across lines or imaging parameters. All mRNA levels for a given experiment are compared within littermates, and labeling and imaging were performed together to minimize technical variation. However, data from each line was collected in completely separate experiments, imaging parameters were optimized separately, and it is of course impossible to compare littermates across different genetic lines, so a rigorous comparison is not possible across these datasets.

d. In line 143, it states that '~10% of *Brn3bcKO* M2/M4 cells stratified to the OFF sublamina.' This observation is very interesting, and 10% is significant. A few representative images should be presented somewhere, ideally with a reference marker such as *Chat*.

We thank the reviewer for raising this important point. In the new version of the manuscript, we added representative traces of M2 and M4 ipRGCs from *Brn3bcKO* mice, with dendritic arbors stratifying in the OFF sublamina of the IPL, in combination with ChAT labeling as a reference marker (Extended Data Figure 10).

Reviewers 2 and 4

Major Comments

1. The authors use TRAPseq to sequence *Opn4Cre* expressing neurons in the retina and illustrate that ipRGC-specific mRNA are up or downregulated when *Brn3b* is deleted. High *Brn3b* expressing neurons (M4-6) show downregulation of specific genes, indicating that *Brn3b* is important for the transcriptional regulation of these genes. Low *Brn3b*-expressing ipRGCs (M1-3) show upregulation of some ipRGC-specific genes, illustrating that *Brn3b* may be involved in the transcriptional suppression of these genes. A mechanistic description of how *Brn3b* might underlie the subtype identity of ipRGCs is unclear, and it is also unclear how the *Brn3b*-dependent regulation of these transcripts impacts cellular diversity or function. At present, there is only a descriptive tie between *Brn3b* deletion and changes in other genes. Given the results in the later parts of the manuscript indicating *Opn4* mRNA and melanopsin protein are upregulated in *Brn3bcKO* mice, it is confusing that *Opn4* gene expression is not included in the TRAPseq analysis.

To address the comment on mechanisms, we analyzed BRN3B Cut & Tag data from E14.5 retina¹ together with our Brn3bcKO TRAP-seq data from adult ipRGCs. We find that **BRN3B binding is enriched at genomic regions overlapping with genes that are downregulated, but not upregulated, upon Brn3bcKO**. We note that this overlap is likely an underestimate because many genes expressed in adult ipRGCs are absent E14.5 retina¹ (comparing scRNA-seq datasets¹¹), and thus BRN3B binding sites for these adult-specific genes are also missing in E14.5 retina. Overall, these findings suggest that Brn3b directly binds target genes for activation to promote the identity of *Brn3b*^{high} M4 ipRGCs, while it negatively regulates other genes enriched in *Brn3b*^{low} M1 ipRGCs via mechanisms independent of BRN3B binding.

Regarding the comment on *Opn4* regulation, we performed new experiments and found that **Brn3bcKO increases *Opn4* expression within the M1, M2, and M4 ipRGC subtypes** (Extended Data Figure 5D and 5G, elaborated on in response to comment 3 below), further supporting an antagonistic link between *Brn3b* and *Opn4* expression. In our TRAPseq analyses using *Opn4*^{Cre/+}; *Brn3b*^{cKOAP}; *Rpl22*^{HA} or *Opn4*^{Cre/+}; *Rpl22*^{HA} mice, we observed that HA immunoprecipitation from Brn3bcKO animals resulted in a modest increase in cone cell transcripts. This suggests that we may be underestimating ipRGC transcripts in the Brn3bcKO animals due to the increase in ratio of cone cells among the capture cells, which could obscure the ~10-20% increase in *Opn4* levels in ipRGCs.

2. Given the OPN4Cre-dependent excision of *Brn3b* occurs “well after post-mitotic ipRGC specification” and there appears to be limited differences in the most pronounced subtype-specific characteristics such as: melanopsin protein concentrations, melanopsin response (limited analysis to M4 ipRGCs), dendritic stratification, central projection targets (only minimally examined with retrograde injections into the SCN), how can the authors rule out the hypothesis that subtype specification (M1-M6) occurs during post-mitotic specification and before their *Brn3b* deletion?

We understand how the wording on this point may have been confusing. Because *Opn4* expression, which defines ipRGCs, must be present prior to excision of BRN3B, it is reasonable to assume that the ipRGCs have become postmitotic and their fate as an RGC has been defined by the time melanopsin expression is detectable. To be clear, we are not trying to make claims about when individual ipRGC subtype specification occurs. Indeed, our data provide support for tuning subtype-defining features, but do not indicate that BRN3B is required post mitotically for determining overall ipRGC subtype identity. We have clarified this language in the text and thank the reviewer for bringing this to our attention.

3. Next, the manuscript examines the *Brn3b* regulation of melanopsin expression in Brn3bcKO mice indicating that Brn3bcKO increases *Opn4* mRNA expression and melanopsin protein levels. These data appear to support the conclusion that Brn3bcKO alters *Opn4* mRNA and melanopsin protein levels, but this is confusing because if the hypothesis is that *Brn3b* suppresses *Opn4* and melanopsin then one would expect that the *Opn4* mRNA and melanopsin protein would be particularly high in the M4-M6 ipRGCs following *Brn3b* deletion, which is not clear from the illustrated data. The example images shown in Fig. 2D seem to illustrate some weak staining of at least one low *Brn3b* ipRGC in the control (weak and large soma in the middle) whereas this is not present in the Brn3bcKO. Which areas of retina are these examples taken from, dorsal or ventral retina? Perhaps these images could be reexamined to determine if the larger number of ipRGCs expressing high levels of melanopsin are putative M1 or M2 ipRGCs and some quantification of their soma size might shed light on this.

We have added new analyses to examine *Opn4* expression levels in ipRGC subtypes. First, by binning soma size in our retinal sections, **we defined putative M1, M2, and M4 subtypes and found increased *Opn4* mRNA levels in all three groups in Brn3bcKO mice compared to control littermates** (Extended Data Figure 5D). Second, we performed new experiments using RNA scope to label for Calbindin and *Opn4* mRNA. This approach allows us to quantify *Opn4* labels in the *Calb1*⁺ *Opn4*⁺ cells, which represent the M4 ipRGCs. We found that ***Opn4* mRNA levels were significantly higher in M4 ipRGCs** (Extended Data Figure 5E-I), supporting our findings using soma size. Third, we now provide additional representative images from our anti-melanopsin immunolabeling from all quadrants of Brn3bcKO and control littermates to show more raw data (Extended Data Fig. 7D).

4. Next the manuscript examines the impact of *Brn3b* expression on the morphology of ipRGCs using Brn3bcKO mice and intravitreal injections of Cre-dependent AAV expressing mCherry to label M2 and M4 ipRGCs and

retrograde injections into the SCN to label M1 ipRGCs. Brn3bcKO clearly has an effect on reducing the dendritic arbor size of ipRGCs, but it is a bit of a stretch to say that reduced arbor size and reduced arbor complexity pushes the M2-M4 ipRGCs towards a M1 morphological phenotype. Some other clarification is needed in this section (lines 136-146) as the authors state that “M1 cells are not labeled at this low titer” but also mention ~10% of ipRGCs stratified in the OFF sublamina. Are these M1 ipRGCs, or are they putative M2 and M4 ipRGCs that have been mis-stratified? Dendritic stratification is one of the most defining characteristics of the M1 ipRGCs. Are these OFF stratifying ipRGCs SMI-32 positive or negative? The results of these data suggest that Brn3b controls aspects of soma and dendritic growth, but it is unclear from the presented data how Brn3b defines the morphological identity of these specific subtypes.

We thank the reviewer for raising this important point. In the new version of the manuscript, we added representative traces of putative M2 (SMI32-negative) and M4 (SMI32-positive) ipRGCs from Brn3bcKO mice, with dendritic arbors stratifying in the OFF sublamina of the IPL, using ChAT as a reference marker (Extended Data Figure 10). Our intention is to highlight that changes in the features move toward the simpler, smaller arbors of M1 ipRGCs. We now note in the discussion that ipRGC subtypes shift features, but each type is still distinguishable and present in similar numbers, indicating that they do not transform in the absence of Brn3b.

5. Next, the manuscript describes the impact of Brn3bcKO on the electrophysiological properties of M4 ipRGCs. M4 ipRGCs in Brn3bcKO mice showed increased input resistance and increased excitability when compared to controls and were more prone to go into depolarization block to high amplitude current injection. The intrinsic photocurrent of M4 ipRGCs however, was unaffected by Brn3bcKO. It is difficult to reason how Brn3b deletion is causing the changes in M4 input resistance and excitability outside of the smaller soma size and dendritic size. The authors accounted for current density but do not detail how the current density was accounted for or calculated. If there is a Brn3b-mediated change in the input resistance and excitability and it does not relate to the changes in morphology, there is no mechanism proposed for how this occurs. Is this due to changes in the density of ion channels? Is there any more evidence to better support this claim?

To account for current density, we normalized the current injections to cell size by dividing by the whole cell capacitance (Extended Data Figure 15). This calculation is now explained in the methods.

The Brn3b-dependent changes in excitability are likely due to changes in the density of ion channels. Many voltage-gated sodium and potassium channels show graded expression along the M1-M4 gradient based on scRNAseq data¹¹. Additionally, based on the TRAPseq data, many voltage-gated ion channels show differential expression in the Brn3bcKO ipRGCs compared to control ipRGCs (Reviewer Figure 1). Likewise, for M4 ipRGCs we saw an increase in input resistance even when normalizing for decreased cell size, though not for M2 or M1 cells, suggesting that different mechanisms or channel types may be involved for this input resistance.

6. Previously, the authors claimed that Brn3b controls the level of *Opn4* mRNA and melanopsin expression, but the previous figures don't address the melanopsin expression in M4 ipRGCs. The rationale for examining the electrophysiological changes in M4 ipRGCs in the Brn3bcKO mice is that M4 ipRGCs have the highest levels of Brn3b expression. Given that the melanopsin expression is not examined in the M4 ipRGCs and that there is no change in the intrinsic photocurrent in M4 ipRGCs with Brn3bcKO doesn't this suggest that Brn3b has little impact on melanopsin expression? The physiological data together with the lack of clear RNA and immunofluorescence data from this cell type does not support this main hypothesis. Would it be possible to examine the melanopsin protein levels in the ipRGCs examined in Figure 3?

As mentioned in point 3 above, we now approach quantification of *Opn4* levels in M4 ipRGCs. We find that *Opn4* mRNA levels are increased in putative M4 ipRGCs based on soma size (Extended Data Figure 5D) and in cells positive for both *Opn4* mRNA and calbindin in whole mount retinas (Extended Data Figure 5E-I), providing support for this conclusion. It is not possible to use the anti-melanopsin antibody to label M4 ipRGCs without enzymatic amplification¹⁷, so it is not possible to rigorously, quantitatively measure melanopsin protein in adult M4 cells.

Regarding the intrinsic response, it is important to note that if the levels of melanopsin in control M4 cells are able to fully saturate the transduction cascade and open all available channels, then expression of additional

melanopsin would not necessarily drive a larger photocurrent through those transduction channels (though it may enhance the sensitivity at very low light).

7. If *Brn3b* deletion increases the melanopsin expression in ipRGCs (M1 and M2?) there should be physiological data to support this by measuring the intrinsic photocurrents in the ipRGCs that have higher melanopsin expression. It is unclear why the mRNA expression and melanopsin protein expression is examined in detail (Figure 2) but not backed up by electrophysiology to illustrate the physiological impact of this expression change in ipRGCs other than M4s.

We appreciate the reviewer's comment. We now include physiological analyses from M1 and M2 ipRGCs, in addition to M4 ipRGCs (Figure 4 and Extended Data Figures 15-18). With the exception of a potentially faster onset of the M2 intrinsic response, we did not observe other major changes, potentially due to the points discussed in point 6 above.

8. Next, the manuscript details the impact of *Brn3bcKO* on various ipRGC-mediated behavioral outcomes. First, they examine the contrast sensitivity of the optokinetic tracking response, which they claim is mediated by M4 ipRGCs. If *Brn3b* reduces the contrast sensitivity of the OKT through differences in M4 function, then the authors should strengthen this finding by measuring the contrast sensitivity of the M4 ipRGCs in the *Brn3bcKO* mouse retinas. The increased excitability and higher input resistance of these cells without any appreciable differences in cellular function until high levels of current injection would predict the opposite effect where lower levels of contrast are more likely to drive spike output from M4 ipRGCs. You would therefore predict that *Brn3bcKO* mice would have higher contrast sensitivity. Could this effect be mediated through impacts on photoreceptors, or through *Brn3b*-deletion in central neurons, many of which express OPN4 in the *OPN4Cre(saha)* mouse line (PMID: 39127043)?

The increased excitability combined with the faster spike adaptation would feasibly lead to decreased contrast sensitivity behaviorally. We now mention this in the Discussion. It is important to remember that this phenotype could arise from more than changes in melanopsin expression, and will be a complex combination of intrinsic properties, any changes in integration of synaptic inputs, and potential changes in synaptic output or connectivity downstream. Additionally, M2, M5, and M6 ipRGCs also project to the dLGN and the role of each subtype in contrast sensitivity is not fully defined, making interpretation of M4 ipRGCs only recordings and their contribution to the behavioral phenotype difficult in isolation. Thus, a thorough analysis of this mechanism is, we feel, beyond the scope of the current study (though a very important question!). Despite difficulty in pinpointing the source of the deficit, we still thought it important to define the behavioral impacts of BRN3B removal from ipRGCs as part of this initial report.

9. Similarly, the impairment of the bright PLR might arise through differential central mechanisms introduced by *Brn3b*-deletion and this result is also a bit confusing. This is because, *Brn3b* deletion is supposed to increase the melanopsin expression in M1 ipRGCs and one might expect that the increased melanopsin expression would enhance the bright light PLR, which is primarily mediated by the intrinsic photosensitivity of M1 ipRGCs. Is there a mechanism proposed for how *Brn3bcKO* might impact the bright PLR through decreased M1 activity in the retina? Recording from M1 ipRGCs in *Brn3bcKO* retinas might provide some insight into how this might occur. Similarly, issues with the direct mechanistic link between ipRGC dysfunction and circadian entrainment are not strong without direct recordings to confirm that *Brn3bcKO* is impacting the physiology or axon targeting of M1 ipRGCs. These complex behavioral effects might arise from the deletion of *Brn3b* from central targets using this *Cre* line (PMID: 38659888).

We now include recordings of physiological properties of the M1 ipRGCs (Figure 4 and Extended Data Figures 15, 17-18). Like M4 ipRGCs, M1 cells show increased excitability and are more easily driven to spike adaptation. This would reduce the dynamic range of signals coming from M1 ipRGCs. We now mention this in the Discussion. Regarding excision in the brain, it's important to know that BRN3B is expressed in only a few brain regions including the superior colliculus and the periaqueductal gray area¹⁸, and these have not been shown to express *Opn4* in the *Opn4^{Cre}* line¹⁹.

Minor comments:

1. The *Opn4Cre(saha)* mouse has ectopic labeling in amacrine cells of the retina potentially confounding TrapSeq experiments (PMID: 39127043). Are there any controls to determine the ssRNAseq data excludes retinal amacrine cells?

To address this comment, we analyzed adult retina scRNAseq data derived from the Nathans lab²⁰ and found that the genes *Tfap2b* and *Slc6a9* are selectively enriched in GABAergic and glycinergic amacrine cells, respectively (Reviewer Figure 2). These findings are consistent with reports that these genes are well-established markers of amacrine cells²¹. However, in our TRAPseq analyses, we find that *Tfap2b* and *Slc6a9* transcripts are significantly depleted after HA immunoprecipitation using adult retina from *Opn4^{Cre}* driven HA-Rpl22 mice (Reviewer Figure 2). These data suggest that we are not capturing amacrine cells using our TRAPseq approach.

2. Many supplemental figures can be combined with main figs of which there are only 4. Figure 3 could be split into two figures, one for morphology and one for physiology. It would be nice to show the embryonic data in a main text figure.

We agree with the Reviewer in this comment. With the new electrophysiology datasets including M1 and M2 ipRGCs, we separated these properties in a new main figure (Figure 4).

3. Was the FISH processing of control and experimental retinas performed at the same time to account for variability between replicates?

Yes, control and experimental retinas were performed at the same time. We added this clarification to the methods section. Thank you for raising this important point.

4. Were the eyes collected at the same time of day given that *Opn4* expression varies across time of day?

Yes, we collected eyes at the same time of day. We added this clarification to the methods section.

5. Why is there Ca²⁺ in the internal solution? 2mM seems quite high.

This calcium internal concentration is typical for recording from ipRGCs^{7,22-26}.

6. Cells were held at -70mV in current clamp. What are the values of the holding current used to keep cells at -70mV and what was their resting membrane potential? This is important because if the cKO m4 ipRGCs have different resting membrane potential values this might impact their excitability.

There is a slight increase in the resting membrane potential of *Brn3bcKO* M4 ipRGCs and a slight decrease in the holding current to start cells at -70 mV. This current was injected at baseline to account for differences in starting Vm across cells, genotypes, and subtypes.

7. Were experimental mice homozygous for *Brn3bcKOAP*? Or were the heterozygous/have a normal *Brn3b* allele?

We appreciate this comment. They were homozygous for *Brn3bcKOAP* (*Brn3b^{cKOAP/cKOAP}*), we clarified this formatting in the new version.

8. It would be useful to report the top 50 differentially expressed genes and associated p-values from *Opn4Cre* vs total retina and control vs *Brn3b* KO experiments.

As suggested, we have included the list of differentially expressed genes and p-values in the new Extended Data Table 1 in the revised manuscript.

9. Contrast levels are not detailed for the contrast sensitivity experiments.

We added this information to the new version (Extended Data Figure 19C).

10. Gene names should be italicized and protein names in all caps to clearly distinguish gene and protein names/labels.

We added this formatting to the new version.

11. Please use melanopsin instead of “Opn4 protein” or “anti-Opn4”. OPN4 should be capitalized if referring to protein but using Opn4 for the gene and melanopsin for the protein is less confusing.

We added this formatting to the new version.

12. Please report the actual genotype instead of “control” in the text. I.e. Opn4Cre^{+/-};Brn3bcKOAP^{-/-} or include a statement that control animals are Opn4Cre^{+/-};Brn3bcKOAP^{-/-} mice.

We added this formatting to the new version.

13. Line 64/65 – How does Fig 1 B/C show graded expression of transcripts?

We have clarified the sentence to indicate that differentially expressed genes identified in Brn3bcKO retinas often also showed *variable* expression across ipRGC subtypes.

14. Line 86 – should Brn3b KO lead to increased Chrna6 expression in only M4-M6?

As we binned the RNAscope data by soma size to analyze *Opn4* mRNA data suggested in point 3 (Extended Data Figure 5D), we estimated the *Chrna6* mRNA levels in putative M4, M2 and M1 ipRGCs. We observed a decreased *Chrna6* mRNA level in putative M4, and M2 Brn3bcKO ipRGCs while no differences were observed in putative M1 ipRGCs (Reviewer Figure 3).

15. Line 103 – genotype of control and experimental mice should be clearly defined in text and figure legend

We clarify this in the new version.

16. Line 107 – which types of ipRGCs are these?

The ATS antibodies (cat# N38 and N39), label M1, M2, and M3 ipRGCs, as shown in Extended Data Fig. 7A. We have added this clarification to the main text of the revised manuscript.

17. Line 162 – ventral M1 ipRGCs are thought to have the highest melanopsin expression (PMID: 23954426 PMID: 28965762) so, one would expect them to have less, not more, Brn3b than dorsal M1s.

We appreciate this comment. Given our hypothesis that *Brn3b* expression is inversely correlated with *Opn4* expression, we analyzed *Opn4* mRNA levels in putative M1 ipRGCs by binning the RNAscope data based on soma size. Our findings show higher *Opn4* mRNA expression in dorsal compared to ventral putative M1 ipRGCs (Reviewer Figure 4). While this result contrasts with the study that examined this, it aligns with our observation of higher *Brn3b* mRNA (and lower *Opn4*) levels in ventral SCN-projecting M1 ipRGCs, supporting the proposed inverse relationship between *Brn3b* and *Opn4* expression.

18. Line 165 – should say extended fig. 8.

We thank the reviewer for detecting this error. We corrected it in the new version.

19. Line 166 – Please clarify this is mRNA and not protein (“modest amounts of Brn3b”).

We appreciate this comment. We clarified this in the new version.

20. Line 467 – Goat serum listed but donkey serum listed in line 539

We appreciate this comment. We corrected this in the new version.

21. Line 705 – What are the established genetic markers used to establish annotate ipRGC subtypes?

We obtained ipRGC subtype and cluster annotations from the original study¹¹, which were described as M1 (C33/C40), M2 (C31), M4 (C43), and an unclassified subtype (C22). Moreover, C7 and C8 were included given their similarity in gene expression profiles to the M1-M4 ipRGC subtypes, as also noted in recent studies^{12,13}. We denote these as M6 (C7) given its enrichment for *Cdh3*¹⁴ and provisionally as M5 (C8) given its enrichment for a combination of PixON transcripts including *Gnas* derived from Patch-seq analyses^{15,16}.

22. Extended data Fig. 7– typos in panel C/D “lengh”

We appreciate this comment. We corrected this in the new version.

23. Line 761 – should "Ventral...higher levels...than...ventral...(change one to dorsal).

We appreciate this comment. We corrected this in the new version.

Reviewer 3

In this manuscript Aranda et al. use a combination of techniques that all support the conclusion that the levels of expression of *Brn3b* mRNA regulate the expression of melanopsin in ipRGCs as well as the differentiation into different ipRGC types. There are 6 ipRGC types that differ in their projection patterns, morphological properties, and melanopsin expression (leading to differences in light response properties). It was originally reported that M1 ipRGCs which have the highest melanopsin expression are *Brn3* negative, while the other ipRGC types have *Brn3b* expression. More recent RNA profiling and verified here, shows that *Brn3b* is expressed in M1 ipRGCs but the levels are decreased compared to the other ipRGC types. The authors show that deletion of *Brn3b* in non-M1 ipRGCs leads to altered transcriptional patterns of genes expressed in ipRGC subtypes including an increase in *OPN4* expression and this is associated with changes in dendritic morphology, and intrinsic properties toward a M1 fate. Showing that *Brn3b* is an important regulator of ipRGC subtype differentiation. Overall, the results are well controlled and compelling and offer mechanistic insight into how changes in gene transcription can regulate subtle changes between cell types. This work will be of high interest to those studying the development of the retina, and those studying how levels of TF regulate subtype specificity.

I have a few questions/comments that should be clarified/addressed that will help me understand the paper better.

1. According to the Broad single cell RGC data base https://singlecell.broadinstitute.org/single_cell/study/SCP509/mouse-retinal-ganglion-cell-adult-atlas-and-optic-nerve-crush-time-series?genes=Opn4%2CPou4f2%2CChrna6%2CZcchc12&cluster=AtlasRGCs&spatialGroups=--&annotation=Cluster--group--cluster&subsample=all&tab=dotplot#study-visualize *Brn3b* is not restricted to ipRGCs, it is expressed in a large number of non-ipRGC types. Does *Brn3b* regulate *Opn4* expression in these types? The authors only remove it from ipRGCs and only assay changes in ipRGCs but in my mind it could/should be repressing *OPN4* in these other types that are not photosensitive.

In our study, we specifically knockout *Brn3b* expression in ipRGCs and test how BRN3B genetically interacts with the factors that induce melanopsin expression in these cells. For example, TBR2 (*Eomes*) expression appears to be required for inducing *Opn4* expression²⁷. We hypothesize that BRN3B fine-tunes *Opn4* expression in some ipRGC subtypes by negatively regulating the functions of TBR2 or other transcription factors enriched in ipRGCs. Consistent with this hypothesis, we found that *Eomes* mRNA expression levels, like *Opn4* mRNA, are increased in *Brn3b*CKO retinas (Extended Data Figure 6). Also of note is that although removal of *Brn3b* from ipRGCs increases *Opn4* expression, there is still considerable variation of *Opn4* levels within individual ipRGC subtypes. This suggests that BRN3B is necessary to fine tune *Opn4* expression levels in concert with other, yet-unknown, regulatory factors. We now include discussion of this point in the last paragraph of the discussion.

Regarding *Opn4* suppression in other RGCs, *Opn4* expression is robustly suppressed in the vast majority of RGCs, as well as in other neuronal and non-neuronal cell types in the retina, brain, and body. Likewise, *Brn3b* is not widely expressed in the brain or even outside of the retinal RGC layer, and **many *Brn3b* negative RGCs do not contain *Opn4***. Together, this suggests that the predominant mechanisms preventing *Opn4* expression in non-ipRGCs are *Brn3b*-independent. Moreover, to rigorously address this question in non-ipRGCs we would require 1) genetic mouse lines with high specificity for specific canonical RGC types, which are not currently available and 2) these markers would have to be expressed at the appropriate developmental stage to excise *Brn3b* after RGC fate has been specified but prior to development of defining morphophysiological features and 3) given the lack of genetic markers and definitive, RGC-type defining morphophysiological features to quantify the impacts of *Brn3b* on features defining that RGC types identity. Thus, though an incredibly interesting question, the technical limitations and broad scope place this question outside the scope of the current study.

2. *Chrna6* and *Zcchc12* expression is not restricted to ipRGC. What happens to *Chrna6* and *Zcchc12* expression when *Brn3b* is removed from non-ipRGCs.

Though this is a very interesting questions, we are only manipulating *Brn3b* in ipRGCs and can therefore only make conclusions about ipRGC subtypes. As mentioned in our response to point 1 above, we currently lack the genetic tools that would allow us to specifically and postmitotically manipulate *Brn3b* in defined subpopulations of non-ipRGC RGC types. Unfortunately, global *Brn3b* knockout before RGCs are specified, results in the loss of ~80% of RGCs as their fates are converted to other retinal cell types, making this experiment technically infeasible. As new Cre lines are developed for RGC subtypes that express *Brn3b*, it will be of great interest to examine the extent to which *Brn3b* plays similar roles in other RGC types.

3. Do *Brn3b* levels decrease by ½ in heterozygous mice? If so I wonder if this changes the numbers of ipRGC types. In other words, does removing ½ of the expression push cells towards the M1 fate?

We appreciate this comment. We performed RNAscope in flat mounts using *Opn4*, *Brn3b* mRNA probes in *Opn4^{Cre/+} Brn3b^{cKOAP/+}* and we observed **decreased *Brn3b*, and increased *Opn4* mRNA levels compared to control (*Opn4^{Cre/+} Brn3b^{+/+}*) mice** (Reviewer Figure 5). Given the results are similar to what is shown in the full knockout and that there are already 24 supplemental figures included with the manuscript, we include it only as a Reviewer Figure. However, we are open to including it in the full manuscript if requested.

4. Currently the evidence for a graded model is that by removing expression leads to an increase in M1 characteristics, but the model also predicts that an increase of *Brn3b* expression in M1 would direct them to a M4-6 fate, or even a non-ipRGC fate. It would strengthen the argument if this was shown.

We agree that this is a very interesting question. In fact, we have generated a new mouse line that overexpresses *Brn3b* in a Cre-dependent manner in the *Rosa* locus (*Brn3b*cOE). However, this mouse line is still in the characterization phase and will be included as part of a full future manuscript. Including the extensive dataset that would be required to rigorously characterize the new line and quantify melanopsin is, we feel, beyond the scope of the current study. However, we do observe, preliminarily, that *BRN3B* overexpression results in decreased melanopsin expression (Reviewer Figure 6) and increased soma size of ipRGCs (not shown), supporting our model. If seeing these data would assuage the reviewer concerns, we would be happy to show a preview of the melanopsin data just for the reviewers.

5. I worry a bit about the use of “gradient” used in abstract and discussion. I am not clear what it means in this context, high, medium and low levels? Can one really quantify protein expression levels based on RNA scope results to differentiate between discrete steps vs. a gradient of expression?

We thank the reviewer for the thoughtful comment. Wherever we have quantified expression of *Opn4* mRNA or protein, we observe no obvious step-wise changes in expression, which we interpret as a continuous variation in *Opn4* expression across ipRGCs. This is also true for published single-cell RNAseq datasets. However, as the reviewer notes, our methods may not conclusively differentiate between these possibilities with single

molecular resolution. In the revised manuscript, we removed the word “gradient” and replaced it with graded expression patterns, which is a more accurate description of the systematic variation of *Brn3b* that occurs across subtypes.

6. The methods also do not determine whether *Brn3b* directly or indirectly regulates *Opn4* expression. Chip seq experiments using *Brn3b* are needed to distinguish between these possibilities. This should be stated in the discussion if this experiment is deemed too much work for this report.

To address this comment, we analyzed BRN3B Cut & Tag data from E14.5 retina¹ together with our *Brn3bcKO* TRAP-seq data from adult ipRGCs. We find that *Brn3b* binding is enriched at genomic regions overlapping with genes that are downregulated, but not upregulated, upon *Brn3bcKO*. We note that this overlap is likely an underestimate because many genes expressed in adult ipRGCs are absent E14.5 retina¹ (comparing scRNA-seq datasets¹¹), and thus *Brn3b* binding sites for these adult-specific genes are also missing in E14.5 retina. Overall, these findings suggest that *Brn3b* directly binds target genes for activation, while it negatively regulates other genes, including *Opn4*, independent of BRN3B binding.

7. There is no mention of how this result reconciles with studies that show that *eomes/Tbr2* is essential for the expression of melanopsin when removed after development. Was *eomes* found to change expression in *Brn3b* mutants or are there two distinct pathways to get to OPN4?

We appreciate this comment. In the new version of the manuscript, we tested *Eomes* mRNA levels in ipRGCs using RNAscope on cross-sectioned retinas. As expected, we observed an increased expression of *Eomes* in *Brn3bcKO* ipRGCs, following a trend similar to that observed for *Opn4* mRNA levels (Extended Data Figure 6).

8. Line 42 could be more clear about the expression “Notably, *Brn3b* expression is present in newly postmitotic ipRGCs and persists into adulthood, suggesting it may play yet unidentified roles in ipRGC development and function” as stated above *Brn3b* is also expressed in a number of non-ipRGCs.

We appreciate this comment. We clarified this in the new version.

9. Line 243 discussion should be modified. The RGC RNA seq atlas of RGCs shows this to be true. “Surprisingly, we find that even SCN-M1 ipRGCs show low levels of *Brn3b* expression that are critical for shaping cellular properties and their role in circadian photoentrainment, highlighting 245 its critical role in defining the full spectrum of ipRGC subtypes.”

Thank you for pointing this out. To highlight the nuance of current understanding, in the Discussion section we have expanded on the fact that the atlas has shown *Brn3b* expression in both M1 types while mouse genetics experiments in *Brn3bDTA* animals have suggested SCN-M1 ipRGCs are “*Brn3b*-negative” M1 ipRGCs (indeed that is the colloquial nomenclature for these M1 cells).

10. Extended data 2: what are the arrows pointing to? Please add in figure legend. Where are the D-V, N-T axes of the retinas?

We appreciate this comment. This was clarified in the legend of Extended Data Figure 2. Unfortunately, we did not track orientation axes in embryos.

11. Can the authors relate the levels of a transcription factor to other instances where levels have been shown to control differentiation? Is this unique to ipRGCs is this a common phenomenon?

Thank you for pointing out the lack of clarity in the language here as we are not intending to make statements about controlling ipRGC differentiation. Given that *Brn3b* is removed after the *Opn4* gene has been turned on in the *Opn4^{Cre}* line, we interpret the presence of melanopsin to mean that the retinal ganglion cell fate of ipRGCs has already been specified. The *iBrn3bcKO* line supports a postmitotic role for shaping at least subtype-defining melanopsin-levels.

Our data do clearly show that ipRGCs do not transform from ipRGCs into a conventional RGC type. Instead, ipRGC subtype-defining morphophysiological and genetic properties are fine tuned based *Brn3b* levels. Signaling and transcription gradients play important roles in cellular differentiation during the early patterning of tissues and among various non-neuronal cell types²⁸⁻³⁰. However, our understanding of how transcription factor levels control the regulation of postmitotic neuron functions among neuronal subtypes remains limited.

References

- 1 Ge, Y. *et al.* Key transcription factors influence the epigenetic landscape to regulate retinal cell differentiation. *Nucleic Acids Res* **51**, 2151-2176, doi:10.1093/nar/gkad026 (2023).
- 2 Kim, H., Kim, M., Im, S. K. & Fang, S. Mouse Cre-LoxP system: general principles to determine tissue-specific roles of target genes. *Lab Anim Res* **34**, 147-159, doi:10.5625/lar.2018.34.4.147 (2018).
- 3 Hosur, V. *et al.* Large-Scale Genome-Wide Optimization and Prediction of the Cre Recombinase System for Precise Genome Manipulation in Mice. *Res Sq*, doi:10.21203/rs.3.rs-4595968/v1 (2024).
- 4 Tian, X. & Zhou, B. Strategies for site-specific recombination with high efficiency and precise spatiotemporal resolution. *J Biol Chem* **296**, 100509, doi:10.1016/j.jbc.2021.100509 (2021).
- 5 Hu, C., Hill, D. D. & Wong, K. Y. Intrinsic physiological properties of the five types of mouse ganglion-cell photoreceptors. *J Neurophysiol* **109**, 1876-1889, doi:10.1152/jn.00579.2012 (2013).
- 6 Lee, S. K. & Schmidt, T. M. Morphological Identification of Melanopsin-Expressing Retinal Ganglion Cell Subtypes in Mice. *Methods Mol Biol* **1753**, 275-287, doi:10.1007/978-1-4939-7720-8_19 (2018).
- 7 Sonoda, T., Okabe, Y. & Schmidt, T. M. Overlapping morphological and functional properties between M4 and M5 intrinsically photosensitive retinal ganglion cells. *J Comp Neurol* **528**, 1028-1040, doi:10.1002/cne.24806 (2020).
- 8 Donocoff, R. S., Teteloshvili, N., Chung, H., Shoulson, R. & Creusot, R. J. Optimization of tamoxifen-induced Cre activity and its effect on immune cell populations. *Sci Rep* **10**, 15244, doi:10.1038/s41598-020-72179-0 (2020).
- 9 Kellogg, C. M. *et al.* Specificity and efficiency of tamoxifen-mediated Cre induction is equivalent regardless of age. *iScience* **26**, 108413, doi:10.1016/j.isci.2023.108413 (2023).
- 10 Chen, M. Y., Zhao, F. L., Chu, W. L., Bai, M. R. & Zhang, D. M. A review of tamoxifen administration regimen optimization for Cre/loxP system in mouse bone study. *Biomed Pharmacother* **165**, 115045, doi:10.1016/j.biopha.2023.115045 (2023).
- 11 Tran, N. M. *et al.* Single-Cell Profiles of Retinal Ganglion Cells Differing in Resilience to Injury Reveal Neuroprotective Genes. *Neuron* **104**, 1039-1055 e1012, doi:10.1016/j.neuron.2019.11.006 (2019).
- 12 Dyer, B., Yu, S. O., Brown, R. L., Lang, R. A. & D'Souza, S. P. Defining spatial nonuniformities of all ipRGC types using an improved Opn4(cre) recombinase mouse line. *Cell Rep Methods* **4**, 100837, doi:10.1016/j.crmeth.2024.100837 (2024).
- 13 Allen, A. E. *et al.* Altered proportions of retinal cell types and distinct visual codes in rodents occupying divergent ecological niches. *Curr Biol*, doi:10.1016/j.cub.2025.02.014 (2025).
- 14 Quattrochi, L. E. *et al.* The M6 cell: A small-field bistratified photosensitive retinal ganglion cell. *J Comp Neurol* **527**, 297-311, doi:10.1002/cne.24556 (2019).
- 15 Goetz, J. *et al.* Unified classification of mouse retinal ganglion cells using function, morphology, and gene expression. *Cell Rep* **40**, 111040, doi:10.1016/j.celrep.2022.111040 (2022).
- 16 Swygart, D., Yu, W. Q., Takeuchi, S., Wong, R. O. L. & Schwartz, G. W. A presynaptic source drives differing levels of surround suppression in two mouse retinal ganglion cell types. *Nat Commun* **15**, 599, doi:10.1038/s41467-024-44851-w (2024).
- 17 Estevez, M. E. *et al.* Form and function of the M4 cell, an intrinsically photosensitive retinal ganglion cell type contributing to geniculocortical vision. *J Neurosci* **32**, 13608-13620, doi:10.1523/JNEUROSCI.1422-12.2012 (2012).
- 18 Lee, H. *et al.* Brn3b regulates the formation of fear-related midbrain circuits and defensive responses to visual threat. *PLoS Biol* **21**, e3002386, doi:10.1371/journal.pbio.3002386 (2023).
- 19 Ecker, J. L. *et al.* Melanopsin-expressing retinal ganglion-cell photoreceptors: cellular diversity and role in pattern vision. *Neuron* **67**, 49-60, doi:10.1016/j.neuron.2010.05.023 (2010).
- 20 Heng, J. S. *et al.* Comprehensive analysis of a mouse model of spontaneous uveoretinitis using single-cell RNA sequencing. *Proc Natl Acad Sci U S A* **116**, 26734-26744, doi:10.1073/pnas.1915571116 (2019).
- 21 Yan, W. *et al.* Mouse Retinal Cell Atlas: Molecular Identification of over Sixty Amacrine Cell Types. *J Neurosci* **40**, 5177-5195, doi:10.1523/JNEUROSCI.0471-20.2020 (2020).
- 22 Sonoda, T. *et al.* A noncanonical inhibitory circuit dampens behavioral sensitivity to light. *Science* **368**, 527-531, doi:10.1126/science.aay3152 (2020).
- 23 Milner, E. S. & Do, M. T. H. A Population Representation of Absolute Light Intensity in the Mammalian Retina. *Cell* **171**, 865-876 e816, doi:10.1016/j.cell.2017.09.005 (2017).

- 24 Contreras, E., Bhoj, J. D., Sonoda, T., Birnbaumer, L. & Schmidt, T. M. Melanopsin activates divergent phototransduction pathways in intrinsically photosensitive retinal ganglion cell subtypes. *Elife* **12**, doi:10.7554/eLife.80749 (2023).
- 25 Lee, S. K., Sonoda, T. & Schmidt, T. M. M1 Intrinsically Photosensitive Retinal Ganglion Cells Integrate Rod and Melanopsin Inputs to Signal in Low Light. *Cell Rep* **29**, 3349-3355 e3342, doi:10.1016/j.celrep.2019.11.024 (2019).
- 26 Sonoda, T., Lee, S. K., Birnbaumer, L. & Schmidt, T. M. Melanopsin Phototransduction Is Repurposed by ipRGC Subtypes to Shape the Function of Distinct Visual Circuits. *Neuron* **99**, 754-767 e754, doi:10.1016/j.neuron.2018.06.032 (2018).
- 27 Abed, S., Reilly, A., Arnold, S. J. & Feldheim, D. A. Adult Expression of Tbr2 Is Required for the Maintenance but Not Survival of Intrinsically Photosensitive Retinal Ganglion Cells. *Front Cell Neurosci* **16**, 826590, doi:10.3389/fncel.2022.826590 (2022).
- 28 Verma, A. *et al.* TCF1 dosage determines cell fate during T cell development. *Sci Adv* **10**, eado5982, doi:10.1126/sciadv.ado5982 (2024).
- 29 Sagner, A. & Briscoe, J. Morphogen interpretation: concentration, time, competence, and signaling dynamics. *Wiley Interdiscip Rev Dev Biol* **6**, doi:10.1002/wdev.271 (2017).
- 30 Hatzistergos, K. E. *et al.* A novel cardiomyogenic role for Isl1(+) neural crest cells in the inflow tract. *Sci Adv* **6**, doi:10.1126/sciadv.aba9950 (2020).

Reviewer's figures

Reviewer Figure 1. Single-cell RNA sequencing¹¹ and TRAP-seq data in ipRGCs. (A-B) mRNA levels of sodium (A) and potassium (B) voltage-gated ion channels. The arrowheads direction on the right panel indicates up- or downregulation in Brn3bcKO vs. control retinas obtained from TRAPseq data. (C) sodium and potassium voltage-gated ion channels mRNA levels in TRAPseq data.

Reviewer Figure 2. (A) Retinal cell types re-analyzed using scRNA-seq data from adult mice²⁰ and visualized with UMAP. The expression levels of well-characterized marker genes for amacrine cells (AC), such as *Tfap2b* and *Slc6a9*²¹, are shown. (B) The amacrine cell markers *Tfap2b* and *Slc6a9* are depleted following HA immunoprecipitation compared to total retina input (N=2-4 biological replicates). (C) *Rbpms2* mRNA expression in retinal progenitor cells (RPC) or retinal ganglion cells (RGCs) and visualized with UMAP plots of snRNA-seq data from E14.5 retina¹. (D) UMAP plots of genes downregulated in adult ipRGCs upon *Brn3bcKO* that harbor BRN3B peaks (left) or those that do not harbor BRN3B peaks from E14.5 retina (right), and visualized as in (C). E14.5 BRN3B peaks are found at genes with shared expression between adult ipRGCs and E14.5 retina, but not at genes with low expression at E14.5.

Reviewer Figure 3. *Chrna6* mRNA levels are significantly decreased in putative M4, M2 ipRGCs of Brn3bcKO mice compared to control littermates. No differences were observed in M1 ipRGCs (n: 22-60 cells/group). Putative M1 ipRGCs were defined by soma size (diameter bins: M4=20.4-21.4; M2=16.9-17.7 and M1=13.7-14.3 μ m). Lines are median values, n.s. (not significant) $P > 0.05$, *** $P < 0.001$, and Mann Whitney U tests.

Reviewer Figure 4. *Opn4* mRNA levels are significantly higher in dorsal M1 ipRGCs compared to ventral M1 ipRGCs (n: 15-44 cells/group). No significant differences in *Brn3b* mRNA levels were observed. Putative M1 ipRGCs were defined by soma size (diameter bin= 13.7-14.3 μm). Lines are median values, n.s. (not significant) $P > 0.05$, **** $P < 0.0001$, and Mann Whitney U tests.

Reviewer Figure 5. Heterozygous removal of Brn3b in ipRGCs ($Opn4^{Cre/+} Brn3b^{cKOAP/+}$ or Brn3bcKO/+) impacts Brn3b (left) and *Opn4* (right) mRNA levels in ipRGCs. Lines are median values, n.s. (not significant) $P > 0.05$, * $P < 0.05$, **** $P < 0.0001$, one-way ANOVA, Tukey multiple comparison test.

Reviewer Figure 6. Brn3bcOE retinas showed increased *Brn3b* (A) and decreased *Opn4* (B) mRNA expression in ipRGCs. Lines are median values, **** $P < 0.0001$, one-way ANOVA, Tukey multiple comparison test.